# Interpretable Concept Bottlenecks to Align Reinforcement Learning Agents

**Quentin Delfosse**[*,1]  **Sebastian Sztwiertnia**[*,1]  **Mark Rothermel**[1]
**Wolfgang Stammer**[1,2]  **Kristian Kersting**[1,2,3,4]
[1]Computer Science Department, TU Darmstadt, Germany
[2]Hessian Center for Artificial Intelligence (hessian.AI), Darmstadt, Germany
[3]Centre for Cognitive Science, TU Darmstadt, Germany
[4]German Research Center for Artificial Intelligence (DFKI), Darmstadt, Germany
`{firstname.lastname}@cs.tu-darmstadt.de`

## Abstract

Goal misalignment, reward sparsity and difficult credit assignment are only a few of the many issues that make it difficult for deep reinforcement learning (RL) agents to learn optimal policies. Unfortunately, the black-box nature of deep neural networks impedes the inclusion of domain experts for inspecting the model and revising suboptimal policies. To this end, we introduce *Successive Concept Bottleneck Agents (SCoBots)*, that integrate consecutive concept bottleneck (CB) layers. In contrast to current CB models, SCoBots do not just represent concepts as properties of individual objects, but also as relations between objects which is crucial for many RL tasks. Our experimental results[2] provide evidence of SCoBots' competitive performances, but also of their potential for domain experts to understand and regularize their behavior. Among other things, SCoBots enabled us to identify a previously unknown misalignment problem in the iconic video game, Pong, and resolve it. Overall, SCoBots thus result in more human-aligned RL agents.

## 1   Introduction

Deep Reinforcement learning (RL) agents are prone to suffer from a variety of issues that hinder them from learning optimal or generalizable policies. Prominent examples are *reward sparsity* [Andrychowicz et al., 2017] and *difficult credit assignment* [Raposo et al., 2021, Wu et al., 2024]. A more pressing issue is the *goal misalignment* problem. It occurs when an agent optimizes a different *side-goal*, aligned with the original *target goal* during training [Koch et al., 2021], but not at test time. Such misalignments can be difficult to identify [di Langosco et al., 2022]. For instance, in this work we discover that such a misalignment can occur in the oldest and most iconic video game, *Pong* (*cf.* Fig. 1). In Pong, the agent's target goal is to catch a ball with its own paddle and to return it passed the enemy's one. The enemy is programmed to constantly follow the ball, thus, agents can learn to focus on the position of the enemy paddle for placing their own, rather than the position of the ball itself. If left unchecked, such *shortcut learning* [Geirhos et al., 2020] may lead to a lack of model generalization and unintuitive failures *e.g.* at deployment time [Zhang et al., 2021].

Recently eXplainable AI (XAI) methods have emerged to detect such shortcut behavior, by identifying the reasons behind a model's decisions [Schramowski et al., 2020, Roy et al., 2022, Saeed and Omlin, 2023]. However, many XAI methods' explanations do not faithfully present the model's

---

[*]Equal contribution
[2]Code available at https://github.com/k4ntz/SCoBots

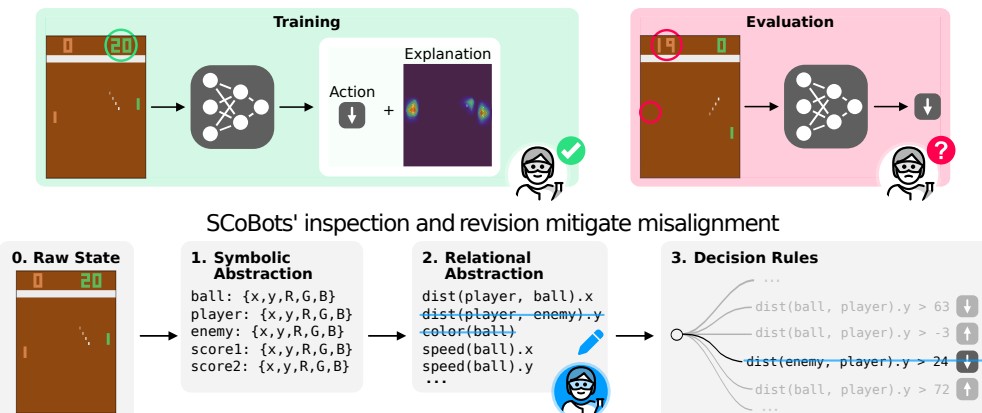

Figure 1: **Successive Concept Bottlenecks Agents (SCoBots) allow for easy inspection and revision.** Top: Deep RL agents trained on Pong produce high playing scores with importance map explanations that suggest sensible underlying reasons for taking an action (✅). However, when the enemy is hidden, the deep RL agent fails to even catch the ball without clear reasons (❓). Bottom: SCoBots, on the other hand, allow for multi-level inspection of the reasoning behind the action selection, *e.g.*, at a relational concept, but also action level. Moreover, they allow users to easily intervene on them (✏️) to prevent the agents from focusing on potentially misleading concepts. In this way, SCoBots can mitigate RL specific caveats like goal misalignment.

underlying decision process [Chan et al., 2022]. For example, importance-map explanations indicate the importance of an input element without indicating *why* this element is important [Kambhampati et al., 2021, Stammer et al., 2021, Teso et al., 2023]. A recent branch of research therefore focuses on models that provide inherent concept-based explanations. Prominent examples are concept bottlenecks models (CBMs), which provide predictive performances on par with standard deep learning approaches for supervised image classification [Koh et al., 2020]. More importantly, CBMs allow to identify and revise incorrect model behavior on a concept level [Stammer et al., 2021].

Contrary to exising CBMs' fields of applications, RL requires relational reasoning [Kaiser et al., 2019]. In this work, we therefore introduce Successive Concept Bottleneck Agents (SCoBots, *cf.* Fig. 1, bottom) bringing the concept bottleneck approach to RL. SCoBots integrate successive concept bottlenecks into their decision processes, where each bottleneck layer provides concept representations that integrate the concept representations of the previous bottleneck layer. Specifically, provided a set of predefined concept functions, SCoBots automatically extract relational concept representations based on objects and their properties of the initial bottleneck layers. Finally, the set of relational and object concept representations are used for optimal action selection. SCoBots thus represent inherently explainable RL agents that, in comparison to deep RL agents, allow for inspecting and revising their learned decision policies at multiple levels of their reasoning processes: from single object properties, through relational concepts to the action selection.

Our evaluations on the iconic Atari Learning Environments (ALE, Mnih et al. [2013]) provides experimental evidence that SCoBots perform on par with deep RL agents. More importantly, we showcase SCoBots' ability to provide valuable explanations and the potential of mitigating a variety of RL specific issues, from reward sparsity to misalignment problems, via simple guidance from domain experts. We identify previously unknown shortcut behavior of deep agents, even on the simple *Pong* game. By utilizing the interaction capabilities of SCoBots, this behavior is easily corrected. Ultimately, our work illustrates the severity of goal misalignment issues in RL and the importance of being able to mitigate these and other RL specific issues, via *relational concept based models*. In summary, our contributions are:
**(i)** We introduce Successive Concept Bottleneck agents (SCoBots).
**(ii)** We show that SCoBots allow to inspect their internal decision processes.
**(iii)** We show that their inherent inspectable nature can helps identifying unknown misalignments.
**(iv)** We show that they allow for human interactions for mitigating various RL specific issues.

We proceed as follows. We introduce SCoBots and discuss their specific properties. We continue with experimental evaluations and analysis. Before concluding, we touch upon related work.

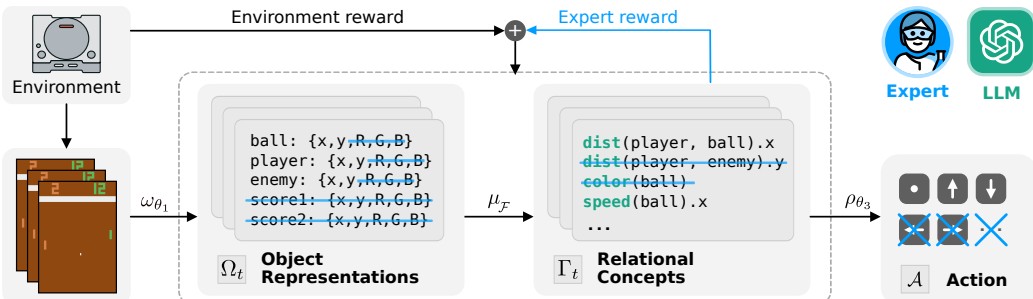

Figure 2: **An overview of Successive Concept Bottlenecks Agents (SCoBots).** SCoBots decompose the policy into consecutive interpretable concept bottlenecks (ICB). Objects and their properties are first extracted from the raw input, human-understandable functions are then employed to derive relational concepts, used to select an action. The understandable concepts enable interactivity. Each bottleneck allows expert users to, *e.g.*, prune or utilize concepts to define additional reward signals.

## 2 Successive Concept Bottleneck Agents

In this work, we represent RL problems through the framework of a Markov Decision Process, $\mathcal{M} =< \mathcal{S}, \mathcal{A}, P_{s,a}, R_{s,a}, \gamma >$, with $\mathcal{S}$ as the state space, $\mathcal{A}$ the set of available actions, $P(s, a)$ the transition probability, and $R(s, a)$ the immediate reward function, obtained from the environment, and $\gamma$ the didscount factor. Classic deep RL policies, $\pi_\theta(s) = P(A = a | S = s)$ parameterized by $\theta$, are usually black-box models that process raw input states, *e.g.* a set of frames, to provide a distribution over the action space [Mnih et al., 2015, van Hasselt et al., 2016].

Concept bottleneck models (CBMs) initiate a learning process by extracting relevant concepts from raw input (*e.g.* image data). These concepts can represent the color or position of a depicted object. Subsequently, these extracted concepts are used for downstream tasks such as image classification. Formally, a bottleneck model $g : x \to c$ transforms an input $x \in \mathbb{R}^D$ with $D$ dimensions into a concept representation $c \in \mathbb{R}^k$ (a vector of $k$ concepts). Next, the predictor network $f : c \to y$ uses this representation to generate the final target output (*e.g.* $y \in \mathbb{R}$ for classification problems).

The main body of research on CBMs focuses on image classification tasks, where the extracted concepts represent attributes of objects. In contrast, RL agents must learn not just from static data, but also through interaction with dynamically evolving environments. RL tasks thus often require relational reasoning, as they involve understanding the relationships between instances that evolve through time and interact with another. This is crucial for learning effective policies in complex, dynamic environments. Note that, in the following descriptions, we use specific fonts to distinguish objects' `properties` from their **`relations`**.

### 2.1 Building inspectable ScoBots

An underlying assumption of SCoBots is that their processing steps should be inspectable and understandable by a human user. This stands in contrast to *e.g.* unsupervised CBM approaches Jabri et al. [2019], Zhou et al. [2019], Srinivas et al. [2020], where there is no guarantee for learning human-aligned concepts. SCoBots rather take a different approach by dividing the concept learning and RL-specific action selection into several inspectable steps that are grounded in human-understandable concept representations. These steps are described hereafter and depicted in Fig. 2.

Similar to other RL agents, SCoBots process the last $n$ observed frames from a sequence of images, $s_t = \{x_i\}_{i=t-n}^t$. For each frame, SCoBots need an initial concept extraction method to extract objects and their properties, as in previous works on CBMs [Stammer et al., 2021, Koh et al., 2020]. Specifically, we start from an initial bottleneck model, $\omega_{\theta_1}(\cdot)$, that is parameterized by parameter set, $\theta_1$. Given a state, $s_t$, this model provides a set of $c_i$ object representations per frame, $\omega_{\theta_1}(s_t) = \Omega_t = \{\{o_i^j\}_{j=1}^{c_i}\}_{i=t-n}^t$, where each object representation corresponds to a tensor of different extracted properties of that object, (*e.g.* its `category`, `position` (*i.e.* $x, y$ coordinates), etc.). As done in previous works on CBMs, the bottleneck model of our SCoBots, $\omega_{\theta_1}$, can correspond to a model that was supervisedly pretrained for extracting the objects and their properties from images.

One of the major differences to previous work on CBMs is that SCoBots further extract and utilize *relational* concepts that are based on the previously extracted objects and their properties. Formally, SCoBots utilize a consecutive bottleneck model, $\mu_{\mathcal{F}}(\cdot)$, called the relation extractor. This model, $\mu$, is parameterized by a predetermined set of transparent relational functions, $\mathcal{F}$, used to extract a set of $d_t$ relational concepts. These relations are based on each individual object (and its properties) for unary relations or on combinations of objects for n-ary relations, and are denoted as $\mu_{\mathcal{F}}(\Omega_t) = \Gamma_t = \{g_t^k\}_{k=1}^{d_t}$. Without strong prior knowledge, $\mathcal{F}$ can initially correspond to universal object relations such as `distance` and `speed`. However, this set can easily be updated on the fly by a human user with *e.g.* additional, novel relational functions. Note that $\mathcal{F}$ can include the identity, to let the relation extractor pass through initial object concepts from $\Omega_t$ to the relational concepts $\Gamma_t$.

Finally, SCoBots employ an action selector, $\rho_{\theta_2}$, parameterized by $\theta_2$, on the relational concepts to select the best action, $a_t$, given the initial state, $s_t$ (*i.e.* the set of frames). Up to now, all extracted concepts in SCoBots, both $\Omega_t$ and $\Gamma_t$, represent human-understandable concept representations. To guarantee understandability also within the action selection step of SCoBots, we need to embed an interpretable action selector. While neural networks, a standard choice in RL literature, are performative and easy to train using differentiable optimization methods, they lack this required interpretability feature. However, Çağlar Aytekin [2022] have recently shown the equivalence between ReLU-based neural networks and decision trees, which in contrast represent inherently interpretable models. To trade-off these issues of flexibility and performance vs interpretability, SCoBots thus break down the action selection process by initially training a small ReLU-based neural network action selector, $\widetilde{\rho}_{\theta_2'}$, via gradient-based RL optimization. After this, $\widetilde{\rho}_{\theta_2'}$ is finally distilled into a decision tree $\rho_{\theta_2}$. Lastly, note that one can add a residual link from the initial object concepts, $\Omega_t$, to the relational concepts, $\Gamma_t$ such that the action selector can, if necessary, also make decisions based on basic object properties, *e.g.*, the height of an object.

Overall, our approach preserves the MDP formulation used by classic deep approaches, but decomposes the classic deep policy $s_t \xrightarrow{\pi_\theta} a_t$ into a successive concept bottleneck one $s_t \xrightarrow{\omega_{\theta_1}} \Omega_t \xrightarrow{\mu_{\mathcal{F}}} \Gamma_t \xrightarrow{\rho_{\theta_2}} a_t$, where $\theta = (\theta_1, \theta_2, \mathcal{F})$ constitutes the set of policy parameters. For simplicity, we will discard the parameter notations in the rest of the manuscript. Further explanations of the input and output space of each module, as well as the properties and relational functions used in this work are provided in the appendix (*cf.* App.A.5 and App. A.6).

Instead of jointly learning object detection, concept extraction, and policy search, SCoBots enable independent optimization of each policy component. Separating the training procedure of different components reduces the complexity of the overall optimization problem [Koh et al., 2020].

## 2.2 Guiding SCoBots

The inspectable nature of SCoBots not only brings the benefit of improved human-understandability, but importantly allows for targeted human-machine interactions. In the following, we describe two guidance approaches for interacting with the latent representations of SCoBots: concept pruning and object-centric rewarding. We refer to the revised SCoBot agents as *guided* SCoBots in the following.

**Concept pruning.** The type and amount of concepts that are required for a successful policy may vary across tasks. For instance, the objects `colors` are irrelevant in Pong but required to distinguish vulnerable ghosts from dangerous ones in MsPacman. However, overloading the action selector with irrelevant concepts, whether these are object properties ($\Omega_t$) or relational concepts ($\Gamma_t$), can lead to difficult optimization (*e.g.* the agent focusing on noise in unimportant states) as well as difficult inspection of the decision tree ($\rho$). Moreover, for a single RL task, the need for specific concepts might even change during training, in *e.g.* progressive environments (where agents need to master early stages before being provided with additional, more complex tasks [Delfosse et al., 2024c]).

SCoBot's human-comprehensible concepts therefore allow domain experts to prune unnecessary concepts. In this way, expert users can guide the learning process towards relevant concepts. Formally, users can (i) select a subset of the object property concepts, $\overline{\Omega}$. Additionally, by (ii) selecting a subset of the relational functions, $\overline{\mathcal{F}}$, or (iii) specifying which objects specific functions should be applied on, experts can implicitly define the relational concepts subset, $\overline{\Gamma}$. Guided SCoBots thus formally refining the policy extraction to $s_t \xrightarrow{\omega_{\theta_1}} \Omega_t \rightarrow \overline{\Omega}_t \xrightarrow{\mu_{\overline{\mathcal{F}}}} \overline{\Gamma}_t \xrightarrow{\rho_{\theta_2}} a_t$. Furthermore, users can (iv) prune out redundant actions resulting in a new action space $\overline{\mathcal{A}}$.

To give brief examples of these four pruning possibilities we refer to Fig. 2, focusing here on the blue subparts. Particularly, in Pong a user can remove objects (*e.g.* scores) or specific object properties (*e.g.* R,G,B values) to obtain $\overline{\Omega_t}$. Second, the color relation, color$(\cdot)$, is irrelevant for successfully playing Pong and can therefore be removed from $\mathcal{F}$. Third, the vertical distance function (dist$(\cdot,\cdot)$.y) can be prevented from being applied to the (player, enemy) input couple. These pruning actions provide SCoBots with $\overline{\Gamma_t}$. Lastly, the only playable actions in Pong are UP and DOWN. To ease the policy search, the available FIRE, LEFT and RIGHT from the base Atari action space might be discarded to obtain $\overline{\mathcal{A}}$, as they are equivalent to NOOP (*i.e.* no action).

Importantly, being able to prune concepts can help mitigate misalignment issues such as those of agents playing Pong (*cf.* Fig. 1), where an agent can base its action selection on the enemy's position instead of the ball's one. Specifically, pruning the enemy position from the distance relation concept enforces SCoBot agents to rely on relevant features for their decisions such as the ball's position, rather than the spurious correlation with the enemy's paddle.

**Object-centric feedback reward**. Reward shaping [Touzet, 1997, Ng et al., 1999] is a standard RL technique that is used to facilitate the agent's learning process by introducing intermediate reward signals, aggregated to the original reward signal. The object-centric and importantly *concept-based* approach of SCoBots allows to easily craft additional reward signals. Formally, one can use the extracted relational concepts to express a new expert reward signal:

$$R^{\text{exp}}(\Gamma_t) := \sum_{g_t \in \Gamma_t} \alpha_{g_t} \cdot g_t, \tag{1}$$

where $R^{\text{exp}} : \Gamma \longrightarrow \mathbb{R}$ and $\Gamma_t = \mu(\omega(s_t))$ is the relational state, extracted by the relation extractor. The coefficient $\alpha_{g_t} \in \mathbb{R}$ is used to penalize or reward the agent proportionally to relational concepts. Our expert reward only relies on the state, as we make use of the concepts extracted from it. However, incorporating the action into the reward computation is straightforward. In practice, this expert reward signal can lead to guiding the agent towards focusing on relevant concepts, but also help smoothing sparse reward signals, which we will discuss further in our evaluations.

For example, one can impose penalties, based on the distance between the agent and objects (*cf.* "expert reward" arrow in Fig. 2), with the intention of incentivizing the agent to maintain close proximity with the ball. As shown in our experimental evaluation, we can use this concept pruning and concept based reward shaping to easily address many RL specific caveats such as reward sparsity, ill-defined objectives, difficult credit assignment, and misalignment (*cf.* App. A.8 for details on each problem).

## 3 Experimental Evaluations

In our evaluations, we investigate several properties and potential benefits of the transparent SCoBot agents. We specifically aim to answer the following research questions:
**(Q1)** Are concept based agents able to learn competitive policies on different RL environments?
**(Q2)** Does the inspectable nature of SCoBots allow to detect issues in their decision processes?
**(Q3)** Can concept-based guidance help mitigate common RL caveats, such as policy misalignments?
**(Q4)** Can SCoBots learn with imperfect object extraction methods?
**(Q5)** How crucial is the the relation extractor for SCoBots performances?

**Experimental setup:** We evaluate SCoBots on 9 Atari games (*cf.* Fig. 3 from the Atari Learning Environments [Bellemare et al., 2012] (by far the most used RL framework (*cf.* App. A.1), as well as the HackAtari modified [Delfosse et al., 2024a] Pong environments, where the enemy is not visible yet active (*NoEnemy*), and where the enemy stops moving after returning the ball (*LazyEnemy*). We provide human normalized scores (following eq. 3) that are averaged over 3 seeds for each agent configuration. We compare our SCoBot agents to deep agents with the classic convolutional network introduced by Mnih et al. [2015], that process a stack of 4 black and white frames and denote these as deep agents in the following. Note, that we evaluate all agents on the latest *v5* version of the environments[3] to prevent overfitting. All agents are trained for 20M frames under the Proximal Policy Optimization algorithm (PPO, [Schulman et al., 2017]), specifically the stable-baseline3 implementation [Raffin et al., 2021] and its default hyperparameters (*cf.* Tab. 2 in App. A.5).

---

[3]This leads to deep agents overall slightly worse than in Schulman et al. [2017], however we obtain similar results when evaluating on the old *v4* versions (*cf.* Tab. 1).

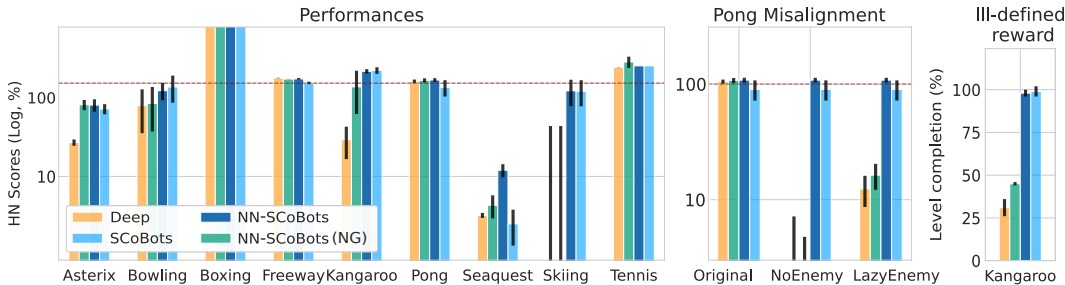

Figure 3: **Object-centric agents can master different Atari environments and interactive SCoBots allow for corrections.** Human-normalized scores of different agents trained using PPO on 9 ALE environments, including deep agents (*i.e.* using CNNs), guided decision tree policy (SCoBots), their neural object-centric baseline (NN-), and these baselines without guidance (NG). SCoBots obtain similar or better scores than the deep agents, showing that object-centric agents can also solve RL tasks while making use of human-understandable concepts (left). Guiding SCoBots allow to correct misalignment in Pong (center) and to obtain the originally intended agents, depicted by a level completion score of 100% on the intended goal's evaluation in Kangaroo (right).

We focus our SCoBot evaluations on the more interesting aspects of the agent's reasoning process underlying the action selection, rather than the basic object identification step (which has been thoroughly investigated in previous works *e.g.* [Koh et al., 2020, Stammer et al., 2021, Wu et al., 2024]). We thus assume access to a pretrained object extractor and provide our agents with object-centric descriptions of these states ($\Omega_t$) based on the information from OCAtari [Delfosse et al., 2024b]. Specifically, in our evaluations, the set of object properties consists of object class (*e.g.* enemy paddle, ball etc.), $(x^t, y^t)$ coordinates, coordinates at the previous position $(x^{t-1}, y^{t-1})$, height and width, the most common RGB values (*i.e.* the most representative color of an object), and object orientation. The specific set of functions, $\mathcal{F}$, that are used to extract object relations is composed of: euclidean distance, directed distances (on $x$ and $y$ axes), speed, velocity, plan intersection, the center (of the shortest line between two objects), and color name (*cf.* App. A.5.1 for implementation details). To distill SCoBots' learned policies, we use the decision-tree extraction algorithm VIPER [Bastani et al., 2018]. For the **human-guided** SCoBot evaluations (denoted as (guided) SCoBots in our evaluations), we illustrate human guidance and prune out the concepts that we consider unimportant to master the game (*cf.* Tab. 4). Furthermore, to mitigate RL specific problems, we also provide SCoBot agents with simulated human-expert feedback signals, the details of which we describe at the relevant sections below. More details about the setup and the hyperparameters are in App. A.5. In our evaluation, SCoBots' training is slightly faster than deep agents' one (*cf.* Appendix A.9).

**SCoBots learn competitive policies (Q1).**
We present human-normalized (HN) scores of both SCoBot and deep agents trained on each investigated game individually in Fig. 3 (left). Numerical values are provided in Tab. 1 (*cf.* App. A.7). We observe that SCoBot agents perform at least on par with deep agents on all games, even slightly outperform these on 5 out of 9 (namely, Asterix, Boxing, Kangaroo, Seaquest and Tennis). Our results suggest that SCoBots provide competitive performances on every tested game despite the constraints of multiple bottlenecks within their architectures. Overall, our experimental evaluations show that RL policies based on interpretable concepts extraction decoupled from the action selection process can, in principle, lead to competitive agents.

**Inspectable SCoBots' to detect misalignments (Q2).**
The main target of developing ScoBots is to obtain competitive, yet *transparent* agents that allow for human users to identify their underlying reasoning process. Particularly, for a specific state, the inspectable bottleneck structure of SCoBots allows to pinpoint not just the object properties, but importantly the relational concepts being used for the final action decision. This is exemplified in Fig. 4 on Skiing, Pong and Kangaroo (*cf.* App. A.4 for explanations of SCoBots on the remaining games). Here we highlight the decision-making path (from SCoBots' decision tree based policies), at specific states of each game. For example, for the game state extracted from Skiing, the SCoBot agent selects RIGHT as the best action, because the signed distance from its character to the left flag is larger than a specific value (+15 pixels). Given the nature of the game this inherent explanation suggests that the agent is indeed basing its selection process on relevant *relational* concepts.



Figure 4: **Interpretable SCoBots allow to follow their decision process**, thanks to their interpretable concepts. The states and associated decision processes of SCoBots (extracted from the decision trees) on Skiing (left), and from unguided SCoBots on Pong (middle) and Kangaroo (right). For example, in this Skiing state, our SCoBot selects RIGHT, as the signed distance between *Player* and the (left) *Flag1* (on the $x$ axis) is bigger than $15$. This agent selects the correct action for the right reasons.

A more striking display of the benefits of the inherent explanations of SCoBot agents is depicted by the Pong agent in Fig 4. The provided explanation suggests that the agent is basing its decision on the vertical positions of the enemy and of its own paddle (distance between the two paddles on the $y$ axis). In fact, this suggests that the agent is largely ignoring the ball's position. Interestingly, upon closer inspection of the enemy movements, we observed that, previously unknown, the enemy is programmed to follow the ball's vertical position (with a small delay). Thus, the vertical positions of these two objects are highly correlated (Pearson's and Spearman's correlations coefficient above $99\%$, *cf.* App. A.7). Moreover, the ball's rendering contains flickering, explaining why SCoBots base their decision on this potentially more reliable feature, the enemy paddle's vertical position.

To validate these findings further, we perform evaluations on 2 modified Pong environments in which (i) the enemy is invisible, yet playing (*NoEnemy*, *cf.* Fig. 1), and on one environment where the enemy is not moving after returning the ball (*LazyEnemy)*. We reevaluate the deep and SCoBot agents that were trained on the original Pong environment on *NoEnemy* and observe catastrophic performance drops in HN scores (*cf.* Fig. 3) at a level of random policies for both types of agents. Evaluations with different DQN agents lead to similar performance drops (*cf.* App. A.7). These drops are particularly striking as initial importance maps produced by PPO and DQN agents highlight all 3 moving objects (*i.e.* the player, ball, and enemy) as relevant for their decisions (*cf.* Fig.1 top left and App. A.7), aligned with findings on importance maps of Weitkamp et al. [2018]. These maps suggest that the deep agents had based its decisions on *right* reasons. Our novel results on the *NoEnemy* and *LazyEnemy* environments, however, greatly calls to question the faithfulness and granularity of such post-hoc explainability methods. They highlight the importance of inherently transparent RL models on the level of concepts, as provided via SCoBots, to identify such potential issues. In the following, we will investigate how to mitigate the discovered issues via SCoBot's bottlenecks.

**Guiding SCoBots to mitigate RL-specific caveats (Q3).**
We here make use of the interactive properties of SCoBots to address several famous RL specific problems: goal misalignment, ill-defined rewards, difficult credit assignment and reward sparsity.

**Realigning SCoBots:** To mitigate the *goal misalignment* issues of the SCoBots trained on Pong, we simply remove (prune) the enemy from the set of considered objects ($\overline{\Omega}$). Thus, the enemy cannot be used for the action selection process. This leads to SCoBots that are able to play Pong and its *NoEnemy* version (*cf.* SCoBot in Fig 3 (center)). Furthermore, this shows that playing Pong without observing the enemy is achievable, by simply returning vertical shots, difficult for the enemy to catch.

**Ill-defined reward:** Defining a reward upfront that incentivizes agents to learn an expected behavior is a difficult problem. An example of *ill-defined reward* (*i.e.* a reward that will lead to an unaligned behavior if the agent maximizes the return) is present in Kangaroo. According to the documentation of the game[4], "the mother kangaroo on the bottom floor tries to reach the top floor where her joey is being held captive by some monkeys". However, punching the monkey enemies gives a higher cumulative reward than climbing up to save the baby. RL agents thus tend to learn to kick monkeys on the bottom floor rather than reaching the joey (*cf.* Fig. 4). For revising such agents, we provide an additional reward signal, based on the distance from the mother kangaroo to the joey (detailed in App. A.8). As can be seen in Fig. 3 (right), this reward allows the guided SCoBots to achieve the originally intended goal by indeed completing the level. In contrast, deep agents and particularly unguided SCoBots achieve relatively high HN scores, but do not complete the levels.

---

[4]`www.retrogames.cz/play_195-Atari2600.php`

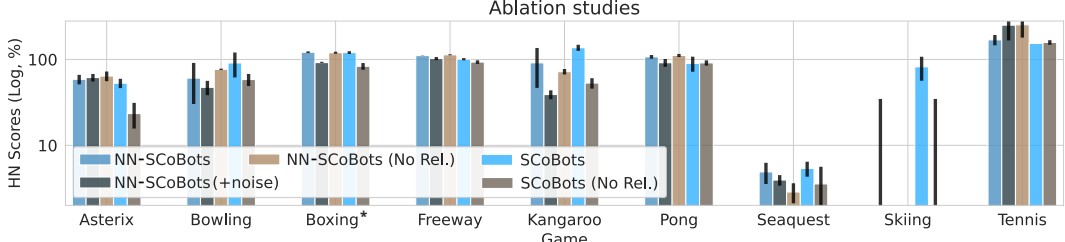

Figure 5: **SCoBots can learn with noisy object detectors, transparent SCoBots rely on relations.** Final human normalized scores (with stds) comparing SCoBots and the object-centric neural baselines (NN-SCoBots), with and without relations. We also provide the scores of NN-SCoBots that learned on noisy environments. The noise only noticeably affects the agents on *Kangaroo*. Ablating the relations is harmless on NN-SCobots, as neural networks can recompute them, but impacts SCoBots performances on 6 games. (*For better visualization, we used a human score of 100 in *Boxing*.)

**Difficult Credit Assignment Problem:** The *difficult credit assignment problem* is the challenge of correctly attributing credit or blame to past actions when the agent's actions have delayed or non-obvious effects on its overall performance. We illustrate this in the context of Skiing, where in standard configurations, agents receive at each step a negative reward that corresponds to the number of seconds elapsed since the last steps (*i.e.* varying between $-3$ and $-7$). This reward aims at punishing agents for going down the slope slowly. Additionally, the game keeps track of the number of flag pairs that the agent has correctly traversed (displayed at the top of the screen, *cf.* Fig. 4), and provide a large reward, proportional to this number, at the end of the episode. Associating this reward signal with the number of passed flags is extremely difficult, without prior knowledge on human skiing competitions. In the next evaluations, we provide SCoBots with another reward at each timestep, proportional to the agent's progression to the flags' `position` to incentivize the agent to pass in between them, as well as a signal rewarding for higher speed:

$$R^{\text{exp}} = \sum_{o\in\{Flag_1, Flag_2\}} D(Player, o)^t - D(Player, o)^{t-1} + V(Player).y. \tag{2}$$

These rewards are further detailed in App. A.8. They allow guided SCoBots to reach human scores, whereas the deep and unguided SCoBots perform worse than random (*cf.* Fig. 3 (left)). Note that providing additional reward signals, as done above, obviously also allows to mitigate *reward sparsity*, as we illustrate on Pong (*cf.* App. A.8.3 for a detailed explanation).

**Object-centric agents can learn with imperfect object extractors (Q4).**
To test object-centric agents' ability to work in more realistic environments, we have tested if they can learn viable policies with imperfect object extraction methods. Based on Delfosse et al. [2023b]'s detection results, we added a $5\%$ misdetection probability and a Gaussian noise (of 0 mean and 3 pixels of standard deviation on each axis). As depicted in Figure 5, NN-SCoBots (*i.e.* neural network based object-centric agents, using relations) learn comparable policies on all games but *Kangaroo*, demonstrating their ability to mitigate suboptimal object extraction methods. Implementing robustness techniques, such as Kalman filters, would help to further push the performances of such agents.

**Transparent SCoBots benefit from explicit relations (Q5).**
While the ablation of the relation extractor only has a significant impact on the neural-based object-centric agents on 1 out of 9 games (*Seaquest*), it deteriorates the decision tree based SCoBots on 6 environments. This is due to the fact that **relations** can be implicitly recomputed within a neural network, but not within decision trees. As shown by Kohler et al. [2024], the use of the **distance** relation (on a specific axis) allows for performing agents with compact decision tree-based policies. Furthermore, even if some relations can be rediscovered and implicitly encoded within the decision trees, the lack of explicit relational representations can reduce the interpretability of the agents, and will prevent the experts from using these within their guidance.

Overall, our experimental evaluations not only show that SCoBots are on par with deep agents in terms of performances, but that their concept-based and inspectable nature allows to identify and revise several important RL specific caveats. Importantly, in the course of our evaluations, we identified a previously unknown and critical misalignment of existing RL agents on the simple and iconic Pong environment, via the previously mentioned properties of SCoBots.

# 4 Limitations

**On the use of OCAtari.** To limit our resource consumption, we made use of the quasi-perfect object extractor ($\omega$) of OCAtari, which efficiently extract objects from the RAM. We added an ablation to simulate potential imperfect detection capabilities of Atari object extraction methods. Such extractors can be optimized using supervised [Redmon et al., 2016, Locatello et al., 2020] or self-supervised [Lin et al., 2020, Delfosse et al., 2023b] object detection methods. This last work showcase that unsupervised object extraction methods can replace OCAtari at test time, however leading to performance drops. Grandien et al. [2024] have further showcase training RL agents with such pretrained object extractors further improve the object-centric RL agents performances.

**The limit of object-centricity.** Other environments require additional information extraction processes. *E.g.* in *MsPacman*, an agent must navigate a *maze*. Extending the concept representations to cover such concepts in maze or platform environments is an important step for future work. Other representations could allow for the integration of *e.g.* path finding methods such as the A* algorithm.

**Training time of SCoBots**. Thanks to the efficient OCAtari object extraction, SCoBots required in average 7.5 hours of training time, while deep agents needed 10.8 hours (*cf.* Appendix A.9). All SCoBots variations require less training time compared to deep agents on environments with few objects (*e.g. Boxing*, *Pong*, *Tennis*). Currently, at every step, the concept bottleneck values are calculated sequentially in a single process on the CPU, leaving significant room for training time improvement by optimizing these computations (bringing them to GPUs). Thus, in environments with many objects, *e.g. Kangaroo*, the training time of unguided SCoBots exceeded that of its deep agent counterpart. Attention on relevant objects could be used to further save computational resources.

# 5 Related Work

The basic idea of **Concept Bottleneck Models (CBMs)** can be found in work as early as Lampert et al. [2009] and Kumar et al. [2009]. A first systematic study of CBMs was delivered by Koh et al. [2020], followed by Stammer et al. [2021], who described CBMs as two-stage models that first computes intermediate representations used for the final task output. Where learning valuable initial object concept representations without strong supervision is still a tricky and open issue for concept-based models [Lage and Doshi-Velez, 2020, Stammer et al., 2022, Sawada and Nakamura, 2022, Marconato et al., 2022, Steinmann et al., 2023, Stammer et al., 2024], receiving relational concept representations in SCoBots is performed automatically via the function set $\mathcal{F}$. Since the initial works on CBMs they have found utilization in several applications. Antognini and Faltings [2021] *e.g.* apply CBMs to text sentiment classification, and Kraus et al. [2024] to time series' analysis. However, these works consider single object concept representations and focus on supervised learning. The ability of users to revise concepts and decisions has also been parallelly shown by Friedrich et al. [2023].

In fact, CBMs have found their way into RL only to a limited extent. Zabounidis et al. [2023] and Grupen et al. [2022] utilized CBMs in a multi-agent setting, where both identify improved interpretability through concept representations while maintaining competitive performance. Zabounidis et al. [2023] further report better training stability and reduced sample complexity. Their extension includes an optional "residual" layer which passes additional, latent information to the action selector part. SCoBots omit such a residual component for the sake of human-understandability, yet offer the flexibility to the user to modify the concept layer via updating $\mathcal{F}$ for improving the model's representations. Similar to how SCoBots invite the user to reuse the high-level concepts to shape the reward signal, Guan et al. [2023] allow the user to design a reward function based on higher-level properties that occur over a period of time. Lastly, SCoBots not only separate state representation learning from policy search, as done by Cuccu et al. [2020], but also enforce object-centricity, putting interpretability requirements on the consecutive feature spaces.

**Explainable RL (XRL)** is an extensively surveyed XAI field [Dazeley et al., 2022, Krajna et al., 2022, Milani et al., 2023] with a wide range of unsolved issues [Vouros, 2022]. Milani et al. [2023] introduce a taxonomy for XRL methods, with: (1) feature importance methods that generate explanations that point out decision-relevant input features, (2) learning process & MDP methods which present which past experience or MDP components affect the policy, and (3) policy-level methods, describing the agent's long-term behavior. Based on this, SCoBots extract relations from low-level features, making high-level information available to explanations and thereby support feature importance methods.

According to Qing et al. [2022]'s categorization of XRL frameworks, SCoBots are "model-explaining" (in contrast to reward-, state-, and task-explaining). Other XRL methods rely on LLM to explain the policy [Luo et al.] decision trees [Fuhrer et al., 2024, Marton et al., 2024], logic [Jiang and Luo, 2019, Kimura et al., 2021, Delfosse et al., 2023a, Sha et al., 2024] or programs [Verma et al., 2018, Trivedi et al., 2021, Cao et al., 2022, Kohler et al., 2024, Wüst et al., 2024] to encode transparent policies. To overcome the potentially unavailable concepts necessary to learn symbolic policies, Shindo et al. [2024] learn a mixture of symbolic and neural policies. Finally, concepts have further been used to derive reward from context using LLMs, as we did for *Kangaroo* and *Skiing* [Kwon et al., 2023, Kaufmann et al., 2024, Wu, 2024, Shen et al., 2024].

The **misalignment problem** is an RL instantiation of the shortcut learning problem, a frequently studied failure mode that has been identified in models and datasets across the spectrum of AI from deep networks [Lapuschkin et al., 2019, Schramowski et al., 2020] to neuro-symbolic [Stammer et al., 2021, Marconato et al., 2023], prototype-based models [Bontempelli et al., 2023] and RL approaches [di Langosco et al., 2022]. A misaligned RL agent, first empirically studied by [Koch et al., 2021] represents a serious issue, especially when it is misaligned to recognized ethical values [Arnold and Kasenberg, 2017] or if the agent has broad capabilities [Ngo, 2022]. Nahian et al. [2021] ethically align agents by introducing a second reward signal. In comparison, SCoBots aid to resolve RL specific issues such as the misalignment problem through inherent interpretability.

# 6   Conclusion

In this work, we have provided evidence for the benefits of concept-based models in RL tasks, specifically for identifying issues such as goal misalignment. Among other things our proposed Successive Concept Bottleneck agents integrate relational concepts into their decision processes. With this, we have exposed previously unknown misalignment problems of deep RL agents in a game as simple as Pong. SCoBot agents allowed us to revise this issue, as well as different RL specific caveats with minimal additional feedback. Our work thus represents an important step in developing *aligned* RL agents, *i.e.* agents that are not just aligned with the underlying task goals, but also with human user's understanding and knowledge. Achieving this is particularly valuable for applying RL agents in real-world settings where ethical and safety considerations are paramount.

Avenues for future research are incorporating a high level action bottleneck [Bacon et al., 2017]. One can also incorporate attention mechanisms into RL agents, as discussed in [Itaya et al., 2021], or use language models (and *e.g.*, the game manuals) to generate the additional reward signal, as done by Wu et al. [2024]. Additionally, we are considering the use of shallower decision trees [Broelemann and Kasneci, 2019]. An interesting research question is how far the task reward signal can aid in learning games object-centric representations [Delfosse et al., 2023b] in the first place.

# Impact statement

Our work aims at developing transparent RL agents, whose decision can be understood and revised to be aligned with the beliefs and values of a human user. We believe that such algorithms are critical to uncover and mitigate potential misalignments of AI systems. A malicious user can, however, utilize such approaches for aligning agents in a harmful way, thereby potentially leading to a negative impact on further users or society as a whole. Even so, the inspectable nature of transparent approaches will allow to identify such potentially harmful misuses, or hidden misalignment.

**Acknowledgments**

This work has benefited from the HMWK projects "The Third Wave of Artificial Intelligence - 3AI", their joint support of the National Research Center for Applied Cybersecurity ATHENE, via the "SenPai: XReLeaS" project. It has also received support of Hessian.AI, as well as the Hessian research priority program LOEWE within the project WhiteBox, and the EU-funded "TANGO" project (EU Horizon 2023, GA No 57100431). The authors also would like to thank Stefan Lichtenstein, Raban Emunds for their help on refactoring the code, and Dwarak Vittal for his initial contributions.

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

# A Appendix

As mentioned in the main body, the appendix contains additional materials and supporting information for the following aspects: further information on the fact that Atari is the most common set of games (A.1), details on the reward sparsity in Pong (A.8.3), detailed numerical results (A.2), learning curves of our RL agents (A.3), the hyperparmeters used in this work (A.5), formal definitions on the SCoBots policies (A.6) and further details on the misalignment problem in Pong.

## A.1 Atari games are most common set of environments

We here show that the Atari games from the Arcade Learning Environment [Bellemare et al., 2012] is the most used benchmark to test reinforcement learning agents.

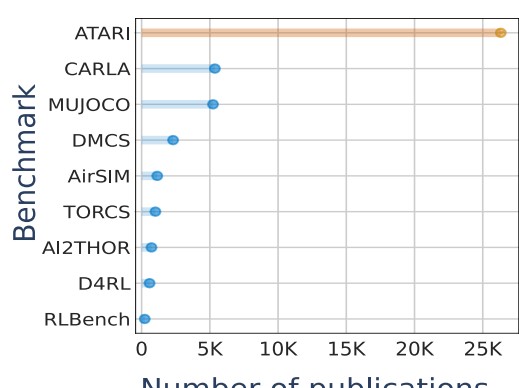

## A.2 Detailed numerical results

For completeness, we here provide the numerical results of the performances obtained by each agent type. For fair comparison, we reimplemeneted the deep PPO agents, and used the default hyperparameters for Atari from stable-baselines3 for the deep RL agents. A detailed overview of hyperparameters and their corresponding values can be found in A.5.

Figure 6: **The Atari Learning Environments is more used in scientific research than the next 8 other benchmarks together**. Graph borrowed from [Delfosse et al., 2024b].

All agents are trained on gymnasium's ALE using 8 actors and 3 training seeds ([0, 16, 32]+rank). Each training seed's performance is evaluated every 500k frames on 4 differently seeded (42+training seed) environments for 8 episodes each. After training, the best performing checkpoint is then ultimately evaluated on 4 seeded (123, 456, 789, 1011) test environments. The final return is determined by averaging the return over 5 episodes per training-test seed and every training seed for the respective environment. We use deterministic actions for both evaluation and testing stages.

| Game | SCoBots-v5 NoGuidance | SCoBots-v5 | PPO-v5 ours | PPO-v4 ours | PPO-v4 Schulman et al. | Random | Human |
|---|---|---|---|---|---|---|---|
| **Asterix** | $5080_{\pm614.8}$ | $5043.3_{\pm729.9}$ | $2126.7_{\pm148.6}$ | $10433_{\pm2773}$ | 4532 | 210 | 8503 |
| **Bowling** | $106.6_{\pm41.8}$ | $137.4_{\pm24.1}$ | $102.2_{\pm39.1}$ | $51.4_{\pm18.0}$ | 40.1 | 23.1 | 160.7 |
| **Boxing** | $97.1_{\pm2.2}$ | $69.0_{\pm10.9}$ | $90.3_{\pm3.0}$ | $99.5_{\pm1.4}$ | 94.6 | 0.10 | 4.3 |
| **Freeway** | $32.8_{\pm0.1}$ | $32.9_{\pm0.7}$ | $33.6_{\pm0.2}$ | $33.6_{\pm0.3}$ | 32.5 | 0.00 | 29.6 |
| **Kangaroo** | $2776.6_{\pm1332.4}$ | $4050.0_{\pm217.8}$ | $790.0_{\pm280.8}$ | $13296.7_{\pm1111.1}$ | 9929 | 52.0 | 3035 |
| **Pong** | $17.2_{\pm1.9}$ | $17.5_{\pm1.8}$ | $16.4_{\pm1.5}$ | $20.9_{\pm0.2}$ | 20.7 | $-20.7$ | 14.6 |
| **Seaquest** | $1055.3_{\pm272.6}$ | $2411.3_{\pm377.0}$ | $837.3_{\pm46.7}$ | $1262.0_{\pm446.1}$ | 1204.5 | 68.4 | 20182 |
| **Skiing** | $-23004_{\pm10333}$ | $-6530_{\pm3326}$ | $-23004_{\pm10333}$ | $-22983_{\pm10333}$ | $-13901$ | $-17098$ | $-4336$ |
| **Tennis** | $2.4_{\pm3.6}$ | $0.0_{\pm0.0}$ | $-0.9_{\pm0.1}$ | $-1.2_{\pm0.3}$ | $-14.8$ | $-23.8$ | $-8.3$ |

Table 1: ScoBots and guided ScoBots obtain similar results than deep agents averaged over 3 seeded runs. Neural refers to the agent that use a CNN instead of our ICB layers. We added the results reported in the original paper Schulman et al. [2017] (original), which are on par with ours, as well as ones from a Random baseline and Humans (from van Hasselt et al. [2016]). Our agents have been trained with only 20M frames.

**Normalisation techniques.** To compute human normalised scores, we used the following equation:

$$\text{score}_{\text{normalised}} = 100 \times \frac{\text{score}_{\text{agent}} - \text{score}_{\text{random}}}{\text{score}_{\text{human}} - \text{score}_{\text{random}}}. \tag{3}$$

## A.3 Learning curves

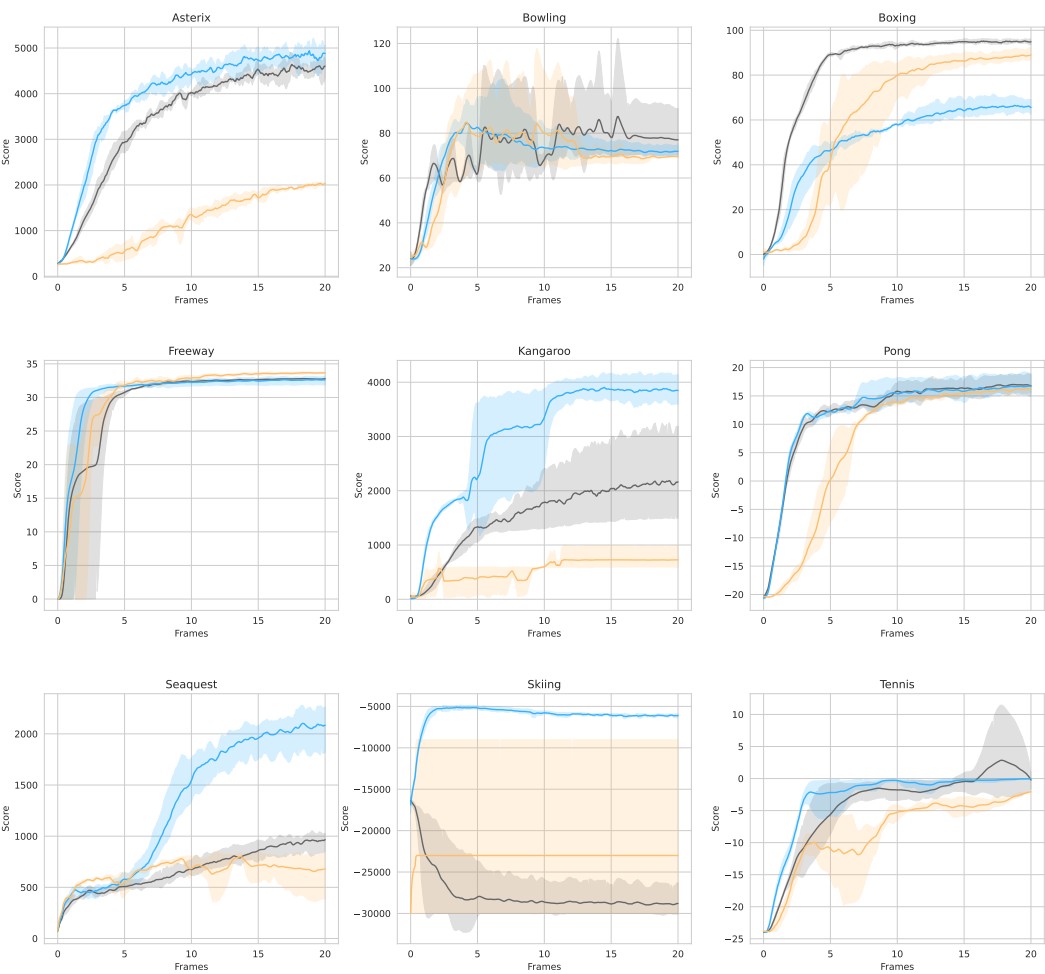

Figure 7: Training Return over frames ($\times 10^6$) seen for ▇ SCoBots, ▇ SCoBots w/o guidance and ▇ neural agents.

## A.4 Agent reasoning through the decision trees

In this section, we provide more decision trees from SCoBots on states from Freeway and Bowling.

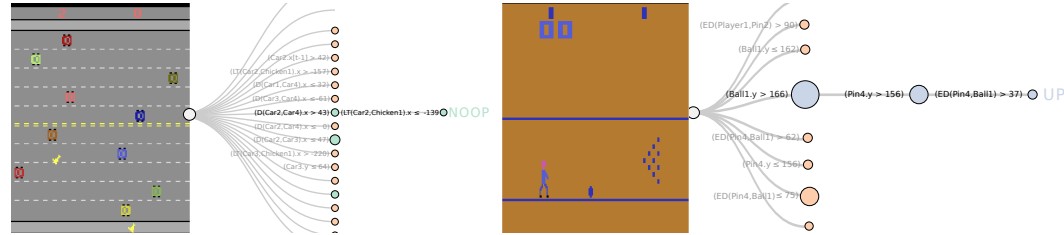

Figure 8: Decision trees from Freeway and Bowling.

## A.5 Hyperparameters and experimental details

All Experiments were run on a AMD Ryzen 7 processor, 64GB of RAM and one NVIDIA GeForce RTX 2080 Ti GPU. Training a SCoBot on 20M frames with 8 actors takes approximately 8 hours. We use the same PPO hyperparameters as the Schulman et al. [2017] agents that learned to master the games. For the Adam optimizer, SCoBots start with a slightly increased learning rate of $1 \times 10^{-3}$ (compared to $2.5 \times 10^{-4}$). The PPO implementation used and the respective MLP hyperparameters are based on stable-baselines3 Raffin et al. [2021]. SCoBots have the same PPO hyperparameter values as deep agents but use MLPs ($2 \times 64$) with ReLU activation functions as policy and value networks. Deep agents use `CnnPolicy` in stable-baselines3 as their policy value network architecture, which aligns with the

| | |
|---|---|
| Actors $N$ | 8 |
| Minibatch size | $32 * 8$ |
| Horizon $T$ | 2048 |
| Num. epochs $K$ | 3 |
| Adam stepsize | $2.5 * 10^{-4} * \alpha$ |
| Discount $\gamma$ | 0.99 |
| GAE parameter $\lambda$ | 0.95 |
| Clipping parameter $\epsilon$ | $0.1 * \alpha$ |
| VF coefficient $c_1$ | 1 |
| Entropy coefficient $c_2$ | 0.01 |

Table 2: PPO Hyperparameter Values. $\alpha$ linearly decreases from 1 to 0 over the course of training.

initial baseline implementation of PPO. Further, we normalize the advantages in our experiments, since it showed only beneficial effects on learning. This is the default setting in the stable-baseline3 implementation.

The Atari environment version used in gymnasium is `NoFrameskip-v4` for agents reproducing the reported PPO results, and `v5` for SCoBots and neural. `v5` defines a deterministic skipping of 5 frames per action taken and sets the probability to repeat the last action taken to 0.25. This is aligned with recommended best practices by Machado et al. [2018]. The experiments using `NoFrameskip-v4` utilize the same environment wrappers as OpenAI's initial baselines implementation[5]. This includes frame skipping of 4 and reward clipping. Frame stacking is not used. SCoBots are not trained on a clipped reward signal. For comparability, the neural agents we compare SCoBots to, are not as well. A list of all hyperparameter values used is provided in Table 2.

### A.5.1 The properties and features used for SCoBots

In this paper, we used different properties and functions (to create features). They are listed hereafter. Further, Table 4 shows a detailed overview of the concepts and actions pruned for every environment evaluated.

## A.6 SCoBots policies: formal definitions

The set $\mathcal{S}$ denotes the problem-specific set of raw environment states as defined by the RL problem, *e.g.*, the space of RGB images $[0, 255]^{512 \times 512 \times 3}$. Similarly, $\mathcal{A}$ represents the action space, *e.g.*, "move right" or "jump."

---

[5]`github.com/openai/baselines`

Table 3:

| | Name | Definition | Description |
|---|---|---|---|
| **Properties** | class | NAME | object class (*e.g.* "Agent", "Ball", "Ghost") |
| | position | $x, y$ | position on the screen |
| | position history | $x_t, y_t, x_{t-1}, y_{t-1}$ | position and past position on the screen |
| | orientation | $o$ | object's orientation if available |
| | RGB | $R, G, B$ | RGB values |
| **Relations** | distance | $D_x(o_1, o_2), D_y(o_1, o_2)$ | distance the $x$ and $y$ axis |
| | euclidean distance | $D_e(o_1, o_2)$ | euclidean distance |
| | trajectory | $landing\_point_x(o_1, o_2)$ | orthogonal projection of $o_1$ onto $o_2$ trajectory |
| | center | $center(o_1, o_2)$ | center of the shortest line between $o_1$ and $o_2$ |
| | speed | $s(x_t, y_t, x_{t-1}, y_{t-1})$ | speed of object (pixel per step) |
| | velocity | $v(x_t, y_t, x_{t-1}, y_{t-1})$ | velocity of object (vector) |
| | color | $color(o_1)$ | the CSS2.1[6] color category (*e.g.* Red) |
| | top k | $Top_k((o_1, ..., o_n), k, class)$ | only $k$ closest ($D_e$ to player) objects visible |

Table 3: Descriptions of properties and relations used by SCoBots.

| | Features | Bowling | Boxing | Pong | Freeway | Tennis | Skiing | Kangaroo | Asterix | Seaquest |
|---|---|---|---|---|---|---|---|---|---|---|
| **Properties** | class | ✓ | ✓ | no enemy | ✓ | ✓ | ✓ | ✓ | ✓ | ✓ |
| | position | ✓ | ✓ | ✓ | ✓ | ✓ | ✓ | ✓ | ✓ | ✓ |
| | position history | ✓ | X | ✓ | ✓ | ✓ | ✓ | ✓ | ✓ | ✓ |
| | orientation | X | X | X | X | X | ✓ | X | X | ✓ |
| | RGB | X | X | X | X | X | X | X | X | X |
| **Relations** | trajectory | X | X | X | X | ✓ | X | X | X | X |
| | distance | ✓ | ✓ | ✓ | ✓ | ✓ | ✓ | ✓ | ✓ | X |
| | euclidean distance | X | X | X | X | X | X | X | X | X |
| | center | X | X | X | X | X | ✓ | X | X | X |
| | speed | X | X | X | ✓ | ✓ | X | X | X | X |
| | velocity | X | X | ✓ | X | X | ✓ | ✓ | ✓ | X |
| | color | X | X | X | X | X | X | X | X | X |
| | k closest objects | 4 | X | X | 4 | X | 2 | 2 | X | X |
| **Actions** | NOOP | ✓ | ✓ | ✓ | ✓ | ✓ | ✓ | ✓ | ✓ | ✓ |
| | FIRE | ✓ | ✓ | ✓ | - | ✓ | - | ✓ | - | ✓ |
| | UP | ✓ | ✓ | - | ✓ | ✓ | - | ✓ | ✓ | ✓ |
| | RIGHT | - | ✓ | ✓ | - | ✓ | ✓ | ✓ | ✓ | ✓ |
| | LEFT | - | ✓ | ✓ | - | ✓ | ✓ | ✓ | ✓ | ✓ |
| | DOWN | ✓ | ✓ | - | ✓ | ✓ | - | ✓ | ✓ | ✓ |
| | UPRIGHT | - | ✓ | - | - | ✓ | - | ✓ | ✓ | - |
| | UPLEFT | - | ✓ | - | - | ✓ | - | ✓ | ✓ | - |
| | DOWNRIGHT | - | ✓ | - | - | ✓ | - | X | ✓ | - |
| | DOWNLEFT | - | ✓ | - | - | ✓ | - | X | ✓ | - |
| | UPFIRE | ✓ | ✓ | - | - | ✓ | - | X | - | - |
| | RIGHTFIRE | - | ✓ | X | - | ✓ | - | X | - | - |
| | LEFTFIRE | - | ✓ | X | - | ✓ | - | X | - | - |
| | DOWNFIRE | ✓ | ✓ | - | - | ✓ | - | X | - | - |
| | UPRIGHTFIRE | - | ✓ | - | - | X | - | X | - | - |
| | UPLEFTFIRE | - | ✓ | - | - | X | - | X | - | - |
| | DOWNRIGHTFIRE | - | ✓ | - | - | X | - | X | - | - |
| | DOWNLEFTFIRE | - | ✓ | - | - | X | - | X | - | - |

Table 4: Feature selection and pruning for guided SCoBots. ✓ denotes the included features, whereas X the features that are pruned out.

Let $O_0$ be the set of all possible objects, identified by their ID and characterized by their properties like position, size, color, etc. Then, $\mathcal{O} := \{O \in \mathcal{P}(O_0) \mid \text{id}(o_1) \neq \text{id}(o_2) \forall o_1, o_2 \in O\}$ is the *object*

$$\mathcal{S} \xrightarrow{\omega} \mathcal{O} \searrow$$
$$\xrightarrow{\mu} \mathcal{R} \xrightarrow{\rho} \mathcal{A}$$
$$\mathcal{F} \nearrow$$

Figure 9: Functional summary of the SCoBots model architecture: $\omega$ maps states to sets of objects, $\mu$ maps sets of objects to relation vectors by applying relational functions, and $\rho$ maps relation vectors to actions.

*detection space*, defined as the family of object sets in which each object (identified by its ID) occurs once at most.

$\mathcal{F}$ is the *relation function space*. It is the family of user-defined relation function sets $F$. Each relation function $f \in F$ is of the form $f : O^k \to \mathbb{R}^n$ for $k, n \in \mathbb{N}$, that is, it maps a fixed number of objects to a real-valued vector. As an instance, the function that maps two objects to the Euclidean distance between each other is a relation function.

The set $\mathcal{R} \subseteq \mathbb{R}^m$ is the *relation* (or *feature*) *space*. The dimension $m$ is implied by $\mathcal{F}$. More specifically, $m = \sum_{f \in \mathcal{F}} \dim(\mathrm{co}(f))$, *i.e.*, $m$ is the sum of each relation function's codomain dimension.

**Definition A.1.** The overall model policy $\pi : \mathcal{S} \to \mathcal{A}$ given relation function set $F$ is defined as $\pi := \rho \circ \mu(\cdot, F) \circ \omega$, where

1. $\omega : \mathcal{S} \to \mathcal{O}$ is the *object detector*, defined as the function that maps a raw state $s \in \mathcal{S}$ to a set of detected objects $O \in \mathcal{O}$.

2. $\mu$ is the *feature selector*
$$\mu : \mathcal{O} \times \mathcal{F} \to \mathcal{R} \tag{4}$$
$$(O, F) \mapsto \mathbf{r} := (f(o_1, ..., o_k))_{f \in F}, \text{ where } o_1, ..., o_k \in O. \tag{5}$$

    That is, $\mu$ applies each relation function $f \in \mathcal{F}$ to the respective detected object(s) in $O \in \mathcal{O}$, resulting in a real-valued relation vector.

3. $\rho : \mathcal{R} \to \mathcal{A}$ is the *action selector* that assigns actions to relation vectors.

See also Figure 9 for a summary. The policy parameters $\theta := (\theta_1, \theta_2, \theta_3)$ split up into the parameters of the object detector, the feature selector, and the action selector, accordingly. They are left out for brevity.

The architecture's bottleneck is induced by the object detector $\omega$ and the feature selector $\mu$. From this function perspective, the bottleneck consists of two stages: an object-centric bottleneck stage at $\mathcal{O}$ and a *semantic* bottleneck stage at $\mathcal{R}$.

### A.7 Pong misalignment problem

The misalignment problem had previously been identified in other games such as the Coinrun platform game [Cobbe et al., 2019], in which an agent's target-goal is to reach a coin which is always placed at the end of a level at training time. When the coin is repositioned at test time di Langosco et al. observed that trained deep RL agents avoid the coin and simply target the end of the level, indicating that agents in fact learn to follow a simpler side-goal rather than the underlying strategy.

For the correlation between the enemy's and the ball's y positions, we let an random agent play the game for 100000 frames and collect the positions of the enemy and ball every 10 frames. We then compute different correlation coefficients, and obtain **99.6%** as Pearson coefficient, **96.4%** as Kendall coefficient and **99.5%** as Spearman coefficient.

We also collected importance maps of deep DQN RL agents playing Pong using ReLU and Rational activation functions (borrowed from Delfosse et al. [2024c].

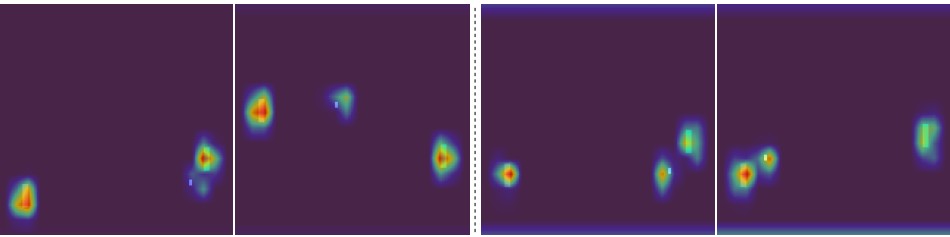

Figure 10: More importance maps of DQN agents playing Pong, with ReLU (left) and rational activation functions (right).

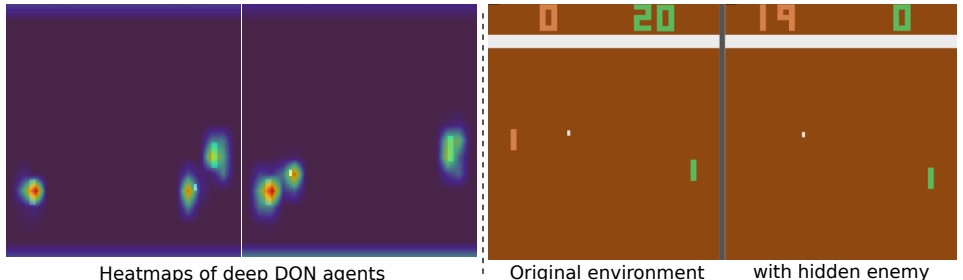

| Heatmaps of deep DQN agents | Original environment | with hidden enemy |

Figure 11: Importance maps of trained deep DQN agents playing Pong (left) show that all 3 moving objects are important for the agent's decision, suggesting that the agent takes aims at sending the ball past the enemy. The same trained agent completely outperform its enemy in the original environment while it cannot return the ball in a modified version of the game where the enemy is hidden.

## A.8 RL specific caveats

In this section, we give more details on the different RL specific caveats, as well

### A.8.1 Ill-defined Objectives

Defining an accurate reward signal when creating an RL task is challenging, as it requires the designer to specify a precise and balanced reward function that effectively guides the agent towards the desired behavior while avoiding unintended consequences [Henderson et al., 2017, Irving et al., 2018]. Shaping the reward after having observed (and understood) the non-intended behavior of the agent is very common [Andrychowicz et al., 2017].

To realign our SCoBots agent, we provide them with a reward signal proportional to the progression to the joey:

```
player = _get_game_objects_by_category(game_objects, ["Player"])

# Get current platform
platform = np.ceil((player.xy[1] - player.h - 16) / 48)  # 0: topmost,
    3: lowest platform

# Encourage moving to the child
if not episode_starts and not last_crashed:
    if platform % 2 == 0:  # even platform, encourage left movement
        reward = - player.dx
    else:  # encourage right movement
        reward = player.dx

    # Encourage upward movement
    reward -= player.dy / 5
else:
    reward = 0
```



Figure 12: iSCoBots learn to land in between the Flag to maximize its reward, when only $R_1^{exp}$ is provided.

### A.8.2 Difficult Credit Assignment

When an outcome is delayed or uncertain, it is challenging to identify the action that is relevant for that outcome [Mesnard et al., 2021]. This problem arises in games like chess, where reward is passed to the agent only when the game is over. The problem can be addressed by more instant informative feedback, given directly after successful actions or after a positive state is reached.

In the main part of this manuscript, we address this problem on the game Skiing. To encourage the agent to go in between the flag, we add a reward that corresponds to the distance to both of the next flags (on the x axis):

$$R_1^{\text{exp}} = D(Player, Flag1).x + D(Player, Flag2).x. \tag{6}$$

This resulted in an agent that learned to stop itself in between the flags (depicted in Fig. 12), showing once again, how difficult it is to create a reward that would favor a specific behavior. To correct it, we therefore adjoined another signal to reward our SCoBot agents proportionally to the `speed` of the agent:

$$R_2^{\text{exp}} = V(Player).y. \tag{7}$$

### A.8.3 The reward sparsity of Pong

Let us move on to the issue of *sparse reward* in the context of RL. In Pong, to score a point, the player needs to return the ball and have it go past the enemy. To do so, the player has to perform a spiky shot, obtained by touching the ball on one of its paddle's sides (otherwise, the shot is flat and the enemy is easily catching it). Its vertical position may vary between $34$ and $190$. Its paddle height is $16$, but it needs to shoot from the side of the paddle (the 3 pixel of each border) to get a spiky shot that is likely to go beyond the opponent. If the balls arrives not close to the top or bottom borders, the enemy will still catch it, which has the probability of $\sim 22\%$ in our experiments. The probability of getting a successful shot is thus of $\sim 6/156 * 0.78 = 3\%$. The average number of corresponding tryouts for the agent to get rewarded is $\sum_{n=1}^{\infty} \left( n \times 0.03 \times 0.97^{n-1} \right) = 33.3$. The agent usually need 60 steps (in average) from the initialization of the point to the reward attribution, hence, it will need $\sim 2200$ steps to be rewarded. This is consistent with the experiment depicted in Figure 13. In this figure, a random agent (original) needs in average 2230 steps to observe reward.

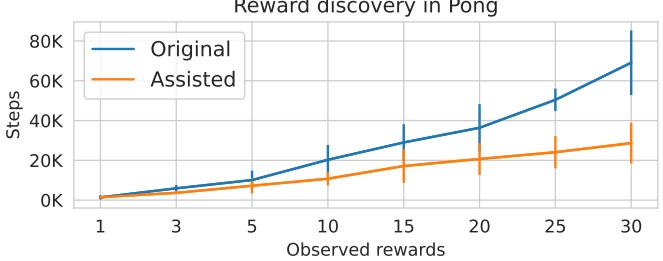

Figure 13: Expert user feedback allow for faster discovering of the reward signal in the sparse reward Pong environment.

Providing a SCoBot agent with an additional reward that is inversely proportional to the distance between its paddle and the ball incentivizes the agent to keep a vertical position close to the ball's

one:

$$R^{\text{exp}} = D(Player, Ball).y \tag{8}$$

This extra reward signal lets a starting agent a winning shot every $\sim 820$, multiplying by $2.7$ their occurrences. Thus, the reward signal in Pong is relatively sparse, but our example illustrates how the interpretable concepts allow to easily guide SCoBots, making up for the reward sparsity. Interestingly, this also giving a clearer incentive to the agent to concentrate on the ball's position, and not on the enemy's one. We observe that such agent do not rely on the enemy to master the game. Alternative techniques such as prioritized experience replay [Schaul et al., 2016] allow offline methods to learn in such environments, but providing a smoother reward signal is another elegant way to address sparsity. Note that sparser environments exists (such as robotics ones).

### A.8.4 Misalignment

In our experimental evaluations, we used the concept based reward shaping to realign the agents on the true (target) task objectives, preventing undesirable behaviors resulting from misaligned optimization. Shortcut learning can be mitigated through the implementation of such reward signals that necessitate the genuine objective's consideration.

### A.9 Computational load

We here provide the overall computational walltime for training each agent type, on each Atari environment. When focusing on a limited number of objects, their computational time is nearly halved, particularly in environments with many objects.

| | SCoBots | Deep | SCoBots (NG) |
|---|---|---|---|
| Asterix | 07 : 08 | 08 : 07 | 12 : 12 |
| Bowling | 07 : 17 | 11 : 13 | 08 : 20 |
| Boxing | 07 : 26 | 11 : 20 | 07 : 32 |
| Freeway | 07 : 34 | 11 : 13 | 09 : 44 |
| Kangaroo | 06 : 01 | 10 : 43 | 21 : 21 |
| Pong | 05 : 42 | 10 : 55 | 07 : 41 |
| Skiing | 07 : 02 | 10 : 37 | 08 : 38 |
| Seaquest | 08 : 53 | 10 : 51 | 25 : 12 |
| Tennis | 11 : 19 | 12 : 08 | 11 : 25 |
| Mean | 07 : 35 | 10 : 47 | 12 : 27 |
| Max | 11 : 19 | 12 : 08 | 25 : 12 |
| Min | 05 : 42 | 08 : 07 | 07 : 32 |

Table 5: **SCoBots (with OCAtari) train faster than deep agents, particularly in environments with a limited number of objects.** Computational training time of each method on each used environment (format HH:MM).

