# OpenReview forum: "Interpretable Concept Bottlenecks to Align Reinforcement Learning Agents"
_NeurIPS.cc/2024/Conference — NeurIPS 2024 poster_

### Official Review · Reviewer_udPa · 2024-06-15

**Soundness:** 3
**Presentation:** 2
**Contribution:** 3
**Rating:** 7
**Confidence:** 2

**Summary:**

This paper provides evidence for the benefits of concept-based models in RL tasks, especially for identifying issues like goal misalignment. The proposed SCoBot integrates relational concepts into decision processes with minimal feedback, which not only brings the advantage of enhanced human understanding (Object Representations) but importantly enables targeted human-machine interactions (Relational Concepts). Experimental results show SCoBots' competitive performances and the potential for domain experts to understand and regularize their behavior.

**Strengths:**

* The method is novel. It not only applies the concept bottleneck to RL tasks, but also further transforms it into an understandable and editable function through LLMs and returns a reward signal.
* The expert reward signal is a reasonable intrinsic reward to solve sparse reward issues.
* The visualization of the experiment makes it easy to understand.

**Weaknesses:**

* Will the space of Object Representations and the types of Relational Concepts change along with the change in the environment? If so, how to solve it?
* Is the concept learned through supervised learning? What if there is no concept label?
* What if the generated relation function keeps having errors by LLMs?

**Questions:**

See above.

**Limitations:**

If the user does not understand the task and cannot define the concepts and the relation functions well.

---

> ### Author Rebuttal · Authors · 2024-08-06
>
> We first want to thank the reviewer for their work and their appreciation of our manuscript, notably for highlighting the novelty of our method and the clarity of our visualizations. We are also adding to our manuscript new ablation studies. The reviewer can find them in the general answer and in the attached PDF.
> Let us address the opportunities for improvement provided by the reviewer:
>
> **How to adapt to new objects types that could require new relations to consider?**
>
> To extend our method to continual RL settings, i.e. if novel objects are detected, the object detector would provide a new object instance. We could incorporate a mechanism to automatically reprompt the LLM/expert to ask them if this novel object type might necessitate novel relations, and train our SCoBots with these novel objects and relations.
>
> **Is the concept learned through supervised learning? What if there is no concept label?**
>
> Our work indeed assumes access to the objects' properties and labels, and/or to the task context. Interpreting the policies would be difficult without human understandable concepts. However, there exists work in the field of predicate discovery (or predicate invention) [1,2,3,4,5]. Such methods can unsupervisely identify useful relations without access to any label. This represents a valuable future work avenue, so we are adding this discussion to our manuscript.
>
> **What if the LLMs generated relation function keeps having errors?**
>
> In our experimental settings, this has not been observed. A simple first solution here would be to use code checkers, and reprompt the LLM if the code checker returns an error. There also exists research works on more complex approaches to ensure that LLMs provide working code [6,7,8]. We believe that these could also be incorporated in SCoBots.
>
> [1] Clarke, Edmund, et al. "Counterexample-guided abstraction refinement." _Computer Aided Verification: 12th International Conference_, 2000.
>
> [2] Athakravi, Duangtida, Krysia Broda, and Alessandra Russo. "Predicate invention in inductive logic programming." _2012 Imperial College Computing Student Workshop_.
>
> [3] Hocquette, Céline, and Stephen H. Muggleton. "Complete bottom-up predicate invention in meta-interpretive learning." _Proceedings of the Twenty-Ninth International Conference on International Joint Conferences on Artificial Intelligence_. 2021.
>
> [4] Silver, Tom, et al. "Predicate invention for bilevel planning." _Proceedings of the AAAI Conference on Artificial Intelligence_. 2023.
>
> [5] Sha, Jingyuan, et al. "EXPIL: Explanatory Predicate Invention for Learning in Games." _arXiv preprint_ (2024).
>
> [6] Liu, Jiawei, et al. "Is your code generated by chatgpt really correct? rigorous evaluation of large language models for code generation." _Advances in Neural Information Processing Systems_ (2024).
>
> [7] Ni, Ansong, et al. "Lever: Learning to verify language-to-code generation with execution." _International Conference on Machine Learning_. PMLR, 2023.
>
> [8] Kwon, Minae, and Sang Michael. "Reward Design with Language Models." _International Conference on Learning Representations (ICLR)_. 2023.

---

> > ### Comment · Reviewer_udPa · 2024-08-09
> >
> > Thanks for the response. I have carefully reviewed the authors' rebuttal and the reviews from other reviewers. It answered my remaining questions sufficiently and I have no further concerns.

---

### Official Review · Reviewer_Zj4A · 2024-06-29

**Soundness:** 3
**Presentation:** 3
**Contribution:** 3
**Rating:** 7
**Confidence:** 5

**Summary:**

The authors introduce SCoBots, an explainable RL approach that first extracts symbols from images, extracts relations between the objects, and then uses them to train a distilled decision tree policy. They demonstrate better performance than naive CNN-based deep RL agents in Atari games.

For full disclosure, I have reviewed this paper before.

**Strengths:**

### Strengths

**Clarity:** Overall the paper is clear and very well written.

**Method:** This method is pretty intuitive and seemingly novel in the RL context.

- It’s also quite an interesting approach to enable explainable RL.
- generating relational features with LLMs is a smart idea to reduce human burden and per-environment tuning

**Performance:** Performance of SCoBots matches or surpasses that of image-based deep RL agents in Atari.

**Experiments:** While I think there’s some issues with the experiments (see weaknesses below), overall the experiments at a high level do a good job of demonstrating advantages of this approach.

**Weaknesses:**

### Weaknesses

**Missing Baselines/Ablations:**

- A baseline/ablation that are missing that actually were addressed in the last time I reviewed this paper but somehow left out of even the appendix of this version:
    - A missing comparison is that of deep rl agents trained with the presumed object detector that SCoBots assumes access to (not the relational extractor, as that’s a component introduced in this method for this setting). This would be an important comparison because object detectors are pretty standard and very easy to give deep RL agents access to (essentially, give it the same state space as SCoBots). **The authors presented an ablation with noisy object detectors last time in the rebuttal, this should probably be highlighted in the paper.**

**Minor issues:**

- Another prior work that also demonstrates being able to have humans intervene on (program) policies given explicitly extracted concepts: Learning to Synthesize Programs as Interpretable and Generalizable Policies, Trivedi et al. 2021. **The authors claimed to have added both works i suggested in the rebuttal last time, they did add one of them (VIPER).**

**Questions:**

See weaknesses

**Limitations:**

Adequately addressed

---

> ### Author Rebuttal · Authors · 2024-08-06
>
> We thank the reviewer, again, for their valuable time and feedback. We are glad that they found our paper _very well written_, our method _quite interesting_ and that our experiments _do a good job of demonstrating advantages of [our] approach_.
> Let us now address the reviewer's concerns.
>
> **Add missing ablation studies.**
>
> Before adding these ablations, we wanted to assess their soundness by training agents with the CNN/VAE based object extractors from [1]. We conducted prechecking experiments training neural SCoBots on Pong and Boxing using their object extraction methods and obtained comparable results to the one with our simulated noise.
>
> Due to time constraints and limited writing space, we excluded these experiments from the initial submission.
> We have summarized our results in the one-page PDF attachment.
> We are of course incorporating these ablation studies into the main body of the manuscript, along with the following new questions, which are addressed in greater details in our manuscript:
>
> **Q4) Can object-centric agents learn with imperfect object extractors ?**
>
> Yes, SCoBots policies can compensate for noisy object extraction, as the added noise only leads to slight performance drops or to no performance issue in our experimental evaluation. We believe that the use of robustness techniques such as Kalman filters can help to stabilize further the performances of such agents.
>
> **Q5) Can SCoBots learn without the relational concept bottleneck ?**
>
> Yes, the neural SCoBots are mostly able to recompute the relations provided to the SCoBots, and obtain comparable performances to the neural SCoBots that use relations. However, the fully interpretable decision tree based SCoBots already suffer some performance loss, and most importantly, agents that do not use relations are more difficult to interpret, thus leading to more difficult misalignment detection and corrections.
>
> [1] Delfosse, et al. "Boosting object representation learning via motion and object continuity." _Joint European Conference on Machine Learning and Knowledge Discovery in Databases_. Cham: Springer Nature Switzerland, 2023.
>
>
> **Missing related work.**
>
> We apologize for this. We apparently have forgotten this related work, in the submitted version of the manuscript. We added it to another paragraph on interpretable/transparent RL methods, placed in our related work section together with further recent publications about interpretable RL.
> Do not hesitate to provide us with any other reference that we might have missed, we will gladly add them.

---

> > ### Comment · Reviewer_Zj4A · 2024-08-09
> >
> > Thanks for the rebuttal! I am already voting accept so I will not change my score.

---

### Official Review · Reviewer_5yYA · 2024-06-30

**Soundness:** 3
**Presentation:** 2
**Contribution:** 2
**Rating:** 4
**Confidence:** 3

**Summary:**

The paper introduces Successive Concept Bottleneck Agents (ScoBots) that utilizes concept bottleneck models to enhance interpretability and decision-making. SCoBots incorporate successive concept bottleneck layers to integrate relational and object-based concepts for action selection. Unlike traditional deep RL agents, SCoBots allow for the inspection and revision of decision policies at various levels, from individual object properties to relational concepts and ultimately to action choices. The authors provide experimental evidence suggesting that SCoBots perform comparably to deep RL agents while offering significant advantages. Notably, SCoBots can justify their actions and address several RL-specific issues such as reward sparsity and goal misalignment through straightforward guidance. During evaluations, SCoBots identified and corrected a previously unknown shortcut behavior in deep RL agents playing Pong, highlighting the potential of this approach to resolve significant challenges in RL through relational concept-based models.

**Strengths:**

1.	This paper is generally straightforward and well-organized. The structure is clear, and most concepts are clearly defined. The major contribution is easy to understand and follow.
2.	The interpretability offered by SCoBots is user-friendly and easy to comprehend. Leveraging object-oriented conceptions, domain experts can conveniently identify and address inherent errors and issues.
3.	The experimental results show the advantages of SCoBots from multiple perspectives (Q1 to Q3). The issues addressed in these experiments are crucial for developing effective decision-making policies in RL. The results are presented clearly and effectively.

**Weaknesses:**

1.	My major concern is about the **environments that the paper studies**. This paper primarily focuses on the application of models to Atari games. This kind of evaluation was popular in explainable RL a few years ago. However, regardless of how promisely the model can interpret Atari games, its practical relevance remains questionable. People gradually realize one cannot depend on an Atari solver to execute any realistic tasks. In today's context, to ensure that reinforcement learning (RL) has a real impact, the RL community has begun to explore tasks related to Embodied AI, such as vision-based manipulation and autonomous driving. These areas are where RL can truly be profitable, offering more than just conceptual benefits. I strongly recommend that the authors investigate these tasks, even within simulators like RoboGen and RoboCasa, which are publicly available. Pursuing research in these environments can ensure that Explainable Artificial Intelligence (XAI) in RL achieves significant, practical impact beyond mere conceptional advantages.
2.	One of the major innovations, **the relation extractor, is not adequately defined in the paper**. It is unclear whether this component is based on a specific model structure, what its objective function for learning is, or if it involves manually designed functions. I briefly scanned the appendix but was unable to find these details. Please clarify these aspects and consider including them in the main body of the paper for better accessibility and understanding.
3.	To write a self-contained paper, the author should **clarify which object extractor is employed by the authors**. Specifically, details the objective of this extractor and describes how the parameter $\omega_{\theta_1}$ is learned from image inputs, whether through self-supervised or unsupervised methods. This information will help readers better understand the methodology and its implementation.
4.	Line 69 contains an incomplete definition of the Markov Decision Process (MDP). It **omits essential components** such as the initial state distribution and the discount factor. Please include these elements to provide a comprehensive description of the MDP.
5.	**Not all state spaces in sequential decision processes can be accurately represented using object-level information**. Such transformations may result in the loss of critical information. Typically, assumptions of an object-oriented MDP are made before conducting these transformations. I recommend that the author explore the literature on object-oriented MDPs, particularly focusing on how objects and their relationships are modeled using graph neural networks. For instance, consider the work by Carlos Diuk, Andre Cohen, and Michael L. Littman titled "An object-oriented representation for efficient reinforcement learning," presented at the Twenty-Fifth International Conference on Machine Learning (ICML 2008). This paper provides valuable insights into object-oriented approaches in reinforcement learning and could greatly enhance the theoretical foundation of your research.
6.	**The set of relations is fixed** before the learning phase, based either on human knowledge or AI-derived insights (as referenced in the Python code on line 100 and limitations). However, this set is not adaptable to new or previously unobserved relations in new or partially observable environments. For instance, both the set of relations and the corresponding action selectors are tailored to specific tasks, such as Pong, and may not generalize well to other contexts.
7.	The introduction of **additional reward signals** (line 169) can significantly **affect the optimal policy in an MDP**. There is no guarantee that the optimal policies before and after the addition of these signals will be identical. These signals can alter the behavior of policies, which in turn changes the interpretations associated with them. Consequently, the **fidelity of the resulting interpretation becomes a critical issue to consider.**

**Questions:**

1. Why not study more realistic tasks like vision-based manipulation?
2. What are the model structure and objective functions for learning the extractors (relation and object ones)?
3. How to deal with the state space that can not be represented by objects (e.g., windy grid world, 3D robot hand)?
4. How to resolve the fidelity issues coming from the added rewards?

**Limitations:**

Yes.

---

> ### Author Rebuttal · Authors · 2024-08-05
>
> We thank the reviewer for their dedicated effort to help us improve our manuscript. Let us address the raised concerns.
>
> **1. Limited evaluation (Atari)**: ALE environments are light, incorporate a diverse set of challenges, and are still heavily used by RL practitioners. E.g., searching "reinforcement learning" together with "robogen" on Scholar, since 2023 leads to 51 results, 6 with "robocasa", and 9330 with "atari".
> Moreover, our evaluations highlight that even these simple tasks already lead to agents that learn spurious correlations, as acknowledged by reviewer Zj4A ("*the experiments do a good job of demonstrating advantages of this approach*"). Specifically SCoBots interpretability allowed us to detect and correct these, as well as other RL specific caveats, (e.g. sparse reward, often encountered in robotic environments).
> Additionally, we evaluate relational reasoning, and ALE games overall include many interacting objects, thus making it a valid choice for evaluations. Robotic environments are indeed more applicable, but they are typically used for learning accurate controls (e.g. the tasks of RoboGen, and Pick and Place, Turning levers, Twisting knobs of RoboCasa).
> Control policies are complementary to our work, and can be used as high level actions together with our SCoBots higher level reasoning system.
> We are indeed planning to extend SCoBots to robotic environments for future work, e.g., for tasks involving relational thinking such as the Composite Tasks from RoboCasa. But this will require much adaptive work. We are adding this discussion to the paper and thank the reviewer for the hint.
>
> **2. Improve the relation extractor definition.**
> We are now extending the definitions of Diuk et al.:
> Using their definitions of classes, $C_i$, explicitly reintroduced in our manuscript.
> The relation extractor augments the object centric space, by applying interpretable expert/LLM provided python functions to every detected object. We extend Diuk et al.'s relation definition from boolean relations (e.g. touch) to real ones (e.g. distance).
> Formally, $r: C_i \times C_j \rightarrow \mathbb{R}$ is a function, defined over the attributes of objects of classes $C_i$ and $C_j$.
> The relation extractor is meant to provide richer interpretable states, from which an action selection is easy to derive (allowing for compact interpretable policies). The functions are here not optimized through an explicit loss, but are pruned at tree extraction time.
> We added these precisions to our manuscript. Thank you helping us improving the clarity of our method.
>
> **3. Which object extractor is employed ?**
> As said in the evaluation (l.205): "We focus our evaluations on the agent’s reasoning process, rather than the object identification". We thus used the imperfect light-weight object extraction of OCAtari. For a fairer comparison to deep RL agents, we have conducted additional evaluations, with noisy object extraction (detailed in the PDF and global response).
>
> **4. Incomplete MDP definition.**
> Thank you for pointing this out, we have completed this definition (l. 69).
>
> **5. Object-centric states are limited.**
> We agree with this remark. We discuss it in the limitation section (cf. l. 320). To address these cases, we can either find suitable neuro-symbolic representations, or combine the symbolic module with a deep one for cases where symbolic reasoning falls short. We updated our limitation section, thank you.
>
> **6. Fixed set of relations.**
> Indeed, we have not touched upon continual RL. However, our setup is interactive (human the loop RL) so this set can be updated by the expert/LLM if the current one is insufficient. E.g. if an interpretable agent performs worse than the deep baseline, or new object classes are observed, the expert/LLM can be asked to extend the set of available functions. This represents an interesting future work opportunity. We have added this discussion.
>
> **7. Additional reward signals can affect the optimal policy.**
> Modifying the MDP may indeed change the optimal policy, but we report the scores using the original reward signal, thus show that guidance allows for better performing policies in the original MDP. Concerning misinterpretation, our SCoBots policies are completely transparent: one can follow the decision tree (that contains relations and properties of each detected uniquely identified object). Contrary to e.g. importance maps, our method does not imply interpretation hypotheses, but each decision at every time step is fully transparent, independently of the training MDP. We are completing our manuscript to avoid confusion.
>
> **Questions**
> 1. Study more realistic tasks like vision-based manipulation?
> See our detailed answer above (1).
>
> 2. What are the model structure and objective functions?
> See our detailed answer above (2).
>
> 3. How to deal with the state space that can not be represented by objects?
> We actually have a broader definition of objects, which could also be referred to as entity. In this manner, the wind can be represented as an entity (with the characteristic direction and speed). Now if the wind is explicitly measurable, e.g., displayed on the rendered state, the object/concept extractor could extract it as well. For interactive properties that are not explicitly rendered (e.g., wind or invisible wall), we can extend our object detector to deduce properties based on the difference between the expected state and the observed state. We have added this discussion to our limitations.
>
> 4. How to resolve the fidelity issues coming from the added rewards?
> SCoBots have a transparent pipeline, from the detected objects, to each evaluated condition in the decision tree. For example, in the Skiing state depicted in Fig. 4, the agent decides to select RIGHT because the distance between the player and the flag (on the x-axis) is bigger than 15. The interpretable decision is thus readable, independent of the MDP.

---

> > ### Author Response · Authors · 2024-08-11
> > **Reminder: The reviewer-author discussion period ends in a couple of days**
> >
> > We would like to express our sincere gratitude to the reviewer for the thorough and constructive feedback. We are confident that our responses adequately address the concerns raised by the reviewer, including the following points:
> >
> > * our experimental evaluation supports our claims, namely that relational concept bottlenecks can be used in competitive agents and help detect and correct many RL specific caveats,
> > * we now discuss the use of SCoBots in robotic environments (notably the one proposed by the reviewers), in our future work section,
> > * we improved the definitions of the object and the relation extractor, expanding the ones of [1] (and thank the reviewer for the reference),
> > * we highlight that our framework can easily be extended to search for new relations when the initial set is not sufficient, notably in continual RL setting,
> > * our agents are fully transparent and thus do not require humans to derive interpretation about the policy and when guided with a relation-based extra reward, we report their scores on the original MDPs.
> >
> > Please kindly let us know if the reviewer has any additional concerns. We are fully committed to resolving any potential issues, should time permit. Again, we thank the reviewer for all the detailed review and the time the reviewer put into helping us to improve our submission.
> >
> > ------
> >
> > [1] Diuk, Carlos, Andre Cohen, and Michael L. Littman. "An object-oriented representation for efficient reinforcement learning." _Proceedings of the 25th international conference on Machine learning_. 2008.

---

> > ### Comment · Reviewer_5yYA · 2024-08-12
> >
> > Thank you for your detailed response, which has deepened my understanding of this work. Having spent years researching interpretable RL and publishing several well-cited papers, I recently engaged more closely with the real industry and recognized a significant gap between XAI, such as XRL, and their practical applications. While the story of XRL focuses on enhancing user experience and trust, it remains unclear which companies or groups are actively utilizing these techniques. The gap between methods used in settings like Atari and their application in real-world production is substantial. Although this work is ok from a scientific exploration standpoint, the practical necessity and application of these findings remain ambiguous.
> >
> > Based on the above points, I am inclined to maintain my rating, but I am not arguing for any final decision.

---

> ### Author Response · Authors · 2024-08-13
> **Most interpretable RL methods are not evaluate on more complicated benchmarks**
>
> We appreciate the reviewer's feedback and agree that the practical application of RL methods in industrial settings is an important direction for future research, and for research in general. However, it is important to recognize that the scope of our paper is to introduce and validate a novel interpretable reinforcement learning method using the most commonly used ALE RL benchmark, that is already enough to support our claims, to demonstrate that deep agents face many caveats, notably learn misalignments, that are easily detectable and
>  fixable using our concept bottlenecks. Contrary to our work, most transparent RL methods are evaluated on even simpler environments (e.g. Cartpole, GridWorld) [1,2,3,4,5,6,7] and are thus not comparable with most RL research and many post hoc explanation methods use Atari or simpler environments [8,9,10,11]. While our method is indeed applicable to other environments, including industrial ones, demonstrating this would require a separate and more extensive study, as we discussed above, with the development and inclusion of options for e.g. robotic tasks. We believe it is somewhat unfair to critic our work based on this, as the gap between research on benchmark datasets and real-world application is a broader challenge facing the entire RL community. Our aim here is to contribute to the foundational science that will ultimately support such real-world applications.
>
> -----
>
> [1] Verma, A., Murali, V., Singh, R., Kohli, P., & Chaudhuri, S. (2018). Programmatically interpretable reinforcement learning. In International Conference on Machine Learning ICML.
>
> [2] Bastani, O., Inala, J. P., & Solar-Lezama, A. (2020). Interpretable, verifiable, and robust reinforcement learning via program synthesis. In International Workshop on Extending Explainable AI Beyond Deep Models and Classifiers.
>
> [3] Shi, W., Huang, G., Song, S., Wang, Z., Lin, T., & Wu, C. (2020). Self-supervised discovering of interpretable features for reinforcement learning. IEEE Transactions on Pattern Analysis and Machine Intelligence.
>
> [4] Hein, D., Udluft, S., & Runkler, T. A. (2018). Interpretable policies for reinforcement learning by genetic programming. Engineering Applications of Artificial Intelligence.
>
> [5] Silva, A., Gombolay, M., Killian, T., Jimenez, I., & Son, S. H. (2020). Optimization methods for interpretable differentiable decision trees applied to reinforcement learning. In International conference on artificial intelligence and statistics.
>
> [6] Roth, A. M., Topin, N., Jamshidi, P., & Veloso, M. (2019). Conservative q-improvement: Reinforcement learning for an interpretable decision-tree policy.
>
> [7] Jiang, Z., & Luo, S. (2019, May). Neural logic reinforcement learning. In International conference on machine learning (pp. 3110-3119). PMLR.
>
> [8] Mott, A., Zoran, D., Chrzanowski, M., Wierstra, D., & Jimenez Rezende, D. (2019). Towards interpretable reinforcement learning using attention augmented agents. Advances in neural information processing systems.
>
> [9] Shi, W., Huang, G., Song, S., Wang, Z., Lin, T., & Wu, C. (2020). Self-supervised discovering of interpretable features for reinforcement learning. IEEE Transactions on Pattern Analysis and Machine Intelligence.
>
> [10] Lyu, D., Yang, F., Liu, B., & Gustafson, S. (2019). SDRL: interpretable and data-efficient deep reinforcement learning leveraging symbolic planning. In Proceedings of the AAAI Conference on Artificial Intelligence.
>
> [11] Liu, G., Schulte, O., Zhu, W., & Li, Q. (2019). Toward interpretable deep reinforcement learning with linear model u-trees. In Machine Learning and Knowledge Discovery in Databases: European Conference, ECML PKDD.

---

### Official Review · Reviewer_KTbS · 2024-07-13

**Soundness:** 3
**Presentation:** 4
**Contribution:** 3
**Rating:** 8
**Confidence:** 4

**Summary:**

This paper proposes Successive Concept Bottleneck Agents that takes an object oriented view on the environment and construct interpretable policies based on objects and associated properties.

**Strengths:**

- The proposed method is well motivated and has strong empirical performance.
- The object oriented idea has been long existed in the RL literature but not until this paper, with slight help of LLM, the idea of object oriented RL unleashed its real power.

**Weaknesses:**

- Would like to see without the help of OCAtari, how the proposed pipeline would be affected by inaccurate object detectors and how this method could work in the wild, e.g., real life robotics domain.
- Instead of distilling neural networks, to achieve better faithfulness and interpretability against the policy network, it would be better to also show results with Calgary Aytekin's NN to DT conversion algo directly.

**Questions:**

- How would you handle more complex relationships like those cooperative behaviors in multi-agent games?
- In POMDP setting, would you allow the relational functions reveal information on previously unobserved properties? If not, how would you plan to handle them?

**Limitations:**

See weakness.

---

> ### Author Rebuttal · Authors · 2024-08-06
>
> We thank the reviewer for their high appreciation of our paper and valuable feedback. We are glad that they found our method *well motivated* and it's *strong empirical performances*. We address the reviewers' raised points hereafter.
>
> **Add an experiment with inaccurate object detectors:**
>
> We first want to stress that the object detectors included in OCAtari are imperfect (in terms of both F-scores and IoU). Nevertheless, we have conducted a novel experiment, with additional noise (of 5% probability of misdetection and gaussian noise on the position of standard deviation of 3 pixels).
>
> The results are in the PDF, together with another ablation study, using agents with no relational concept bottleneck. \
>
>
> While the results show that the policy is still mainly able to maintain correct performances besides the added noise, Kalman Filters could further help stabilize the policies. We are interested in robotic domains, we plan on expanding our work to integrate detection methods such as Segment Anything methods (maybe with Kalman Filters). We think that this will require adaptations like the use of control policies as options (i.e. high level actions), as we have added in our future work paragraph.
>
>  \
> This suggestion experiment indeed helped us to show that concept bottleneck agents can be used in more complicated environments.
>
> **Use Calgary Aytekin's NN to DT conversion algorithm:**
>
> We have not been able to reproduce this algorithm to directly extract decision trees. We, however, believe that the decision tree that it will provide will be larger than those of VIPER, which reduces the sampling space. As shown by the authors, the decision tree obtained on the simple two moons dataset already contains many nodes. If the reviewer knows of any other implementation (than [the one on this github page](https://github.com/CaglarAytekin/)), we would be happy to try another algorithm again.
>
> We have also tried the REMIX algorithm for rule extraction, but they both led to bad performing agents.
>
> We can add this comment in the paper.
>
>
> ### Questions:
>
>
>
> 1. How would you handle more complex relationships like those of cooperative behaviors in multi-agent games? \
> We have asked an LLM to provide such functions, simply prompting it with: \
> "_What about more complex relationships like those of cooperative behaviors in multi-agent games ? Can you think of any function that might be required for these cases ?_"  \
> Its answer provided python functions to evaluate and perform:
> * _Communication Signal_,
> * _Collective Goal Achievement_,
> * _Formation Maintenance_,
> * _Role Assignment_,
> * _Tactical Retreat or Advance_,
> * _Resource Allocation Among Agents_,
> * _Cooperative Targeting_,
> which showcase abilities to use LLM in Multi Agents RL as well.
> We believe that when provided with more context about the task's objectives, the relational functions will be even more accurate. We can as well add this to the appendix.
> 2. In POMDP setting, would you allow the relational functions to reveal information on previously unobserved properties? If not, how would you plan to handle them?
>
>     Many approaches can be taken. In POMDP, one could indeed maintain a state of beliefs at the relational level as well, from which one could deduce partially occluded properties. For example, an occlusion relation could be utilized, allowing to deduce if an object is likely to be situated behind another one. We are now discussing this advantage in our manuscript. Thank you for the hint!

---

> > ### Comment · Reviewer_KTbS · 2024-08-13
> >
> > Thanks for the reply! Most of my concerns are solved. Two quick follow ups regarding the response:
> > 1. "We have not been able to reproduce this algorithm to directly extract decision trees." Could you be more specific on why the author's code (as in the link you posted) cannot work?
> >
> > 2. My concern is more about revealing "forbidden" attributes to the agents violating the env assumptions of **PO**MDP. If you use the relation function "occlusion" and it tells agent that B is behind A, this feels a little bit cheating to me as it's not sth available in direct image observations.
> >
> > Overall I am satisfied with the paper's contributions and will keep the score. But I am keen to know the detailed answers to my above questions.

---

> > > ### Author Response · Authors · 2024-08-13
> > >
> > > Dear reviewer, thank you for getting back at us. We are happy that we were able to address your concerns and that you are satisfied with our contribution. Concerning your questions:
> > >
> > > 1. We have looked at the NN to DT github repo at this link (https://github.com/CaglarAytekin/NN_DT), which contains only one script file (_yx2reproduce.py_), without any extraction method, but with only one hard coded tree, probably extracted from a TensorFlow based neural network. We were thus not able to adapt any tree extraction algorithm, as none was provided.
> > >
> > > 2. By default, in POMDP settings, we would of course not allow for revealing hidden attributes per se, as this would violate the POMDP assumption. However, we believe that SCoBots could be extended such that the agent can learn at training time that e.g. objects can be occluded. If the agent observes that the object was present in the environments (with low probability of suddenly disappearing) in the previous states, it could make use of the relational functions to update its belief on the raw property of the object to place it behind a potentially occluding object. This would of course be an extension to SCoBots that maintain e.g. a belief, and where this belief about objects properties can be influenced (backward) by the belief about the relational state.
> > >
> > > We hope that this helps clarify your question.
> > >
> > > Thank you again for your insightful comment, that allowed us to consider further potentially interested lines of research.

---

### Author Rebuttal · Authors · 2024-08-06

We first thank all the reviewers for their time and valuable feedback.

The reviews are highly appreciated as they raise very interesting points, which ultimately enable us to further improve our manuscript.

Reviewers **KTbS**, **Zj4A** and **udPa** point out the beneficial and interesting role of LLMs for handling relational functions.

This is great feedback for us as we are very excited about this LLM integration.

Reviewer **5yYA** raised several points concerning formal and methodological clarity, which was very helpful as we strive to make our contribution as clear and as coherent as possible.

In addition, we appreciate this reviewer's feedback of extending the context ScoBots are applied to with the highly interesting context of robotics and its implications.

The clear presentation and comprehensive evaluation are very important to us, therefore we appreciate the positive feedback expressed by all four reviewers in this regard.

As requested, we are providing two additional experiments that are summarized in Figure 1.

The first experiment measures SCoBots' robustness to learn with noisy object extraction methods.

The second one ablates SCoBots' relational concepts bottleneck. These allow us to answer the following two additional questions in our manuscript, here concisely answered:

**Q4**: Can object-centric agents learn with imperfect object extractors?

Yes, SCoBots agents are able to learn using imperfect object extractors. Using extra noise (5% probability of object misdetection and Gaussian noise, with standard deviation of 3px on the x, y position) on the imperfect OCAtari detectors, SCoBots are still able to learn Robust policies only leads to small performance reduction in 4 environments, and no noticeable one in the 5 others. We believe that the incorporation of stabilization techniques (such as Kalman filters) can further help to reduce the performance reduction.

**Q5**: Can SCoBots learn without the relational concept bottleneck?

The neural versions of SCoBots (i.e. SCoBots before decision tree extraction) can implicitly learn to reconstruct most relations within their neural network, we thus observe similar performances to the baseline. However their interpretable versions (that use decision) show performance drops on 7 out of 9 environments. Moreover, this ablation reduces the interpretability and ease of correcting misbehavior.

---

### Comment · Area_Chair_VEPe · 2024-08-12
**Discussion Period almost over**

The discussion period is almost over, so both authors and reviewers please respond to any unaddressed questions. Reviewers, be sure that you have all of the information you need from the authors, since after the 13th, they won't be able to respond.

---

### Decision · Program_Chairs · 2024-09-25

**Decision:**

Accept (poster)

**Comment:**

Reviewers generally agreed that this paper is clear, well written and novel. After some improvements by the authors, the primary weakness identified by any of the reviewers is the difficulty of the benchmark involved (Atari). While I agree that striving for more realistic benchmarks should always be encouraged, considering the novelty of this paper, I believe more realistic benchmarks can be left for future work.